# Less Token, More Signal: MoE Expert Pruning via Critical Token Selection

Zeliang Zong [* 1]   Kai Zhang [* 2 1]   Yarong Wang [3]   Wenming Tan [1]   Ye Ren [1]   Jilin Hu [2]

## Abstract

Mixture-of-Experts (MoE) architectures provide strong scalability for large language models, but their large expert parameter footprint poses challenges for efficient deployment. Expert pruning is widely used to reduce model size and inference cost; however, existing approaches are token-agnostic, treating all tokens equally when estimating expert importance. This uniform treatment dilutes the contributions of informative tokens and leads to suboptimal pruning decisions. To address this fundamental limitation, we propose STEP (Selective Token-guided Expert Pruning), a token-aware framework that rethinks expert pruning from the perspective of selective token guidance. By incorporating loss-aware expert evaluation and a lightweight knowledge-preserving mechanism, STEP reduces information loss while removing redundant experts. Extensive experiments across different MoE architectures and model scales demonstrate the effectiveness of STEP. On the 30B Qwen3 MoE model with 50% expert sparsity, STEP achieves nearly a 50% reduction in memory usage with minimal performance degradation, delivers a $1.5\times$ throughput improvement and completes the entire pruning process within 10 minutes. Codes are available in https://github.com/hikvision-research/STEP.

## 1. Introduction

The exponential scaling of large language models (LLMs) (Brown et al., 2020; OpenAI et al., 2024) has fundamentally transformed natural language processing, with Mixture-of-Experts (MoE) architectures (Jacobs et al., 1991; Roller et al., 2021) emerging as a dominant paradigm for achieving exceptional performance while maintaining computational efficiency. Through sparse expert activation, contemporary MoE implementations optimize the performance-efficiency trade-off remarkably well. For instance, Qwen3-30B-A3B (abbreviated as Qwen3-30A3B) achieves performance parity with the dense Qwen3-14B model while activating only 3B parameters per token from its 30B parameter set (Yang et al., 2025). However, despite achieving computational efficiency during inference, MoE architectures face fundamental deployment bottlenecks due to massive memory footprints (Yi et al., 2025; Liu et al., 2025b). The Qwen3-30A3B model requires over twice the memory capacity of the 14B dense model for parameter storage, creating substantial barriers in resource-constrained environments. This challenge becomes particularly acute for state-of-the-art models: deploying a 671B parameter model (DeepSeek-AI et al., 2025) demands over 1.3TB of GPU memory, rendering it inaccessible to most practitioners.

Given the infeasibility of full retraining due to proprietary training data and extraordinary computational costs, post-training compression has become essential. Current approaches fall into two categories: **expert matrix compression** methods that compress individual expert parameters (Li et al., 2025b; Chen et al., 2025b), and **expert-level pruning** methods that reduce expert count through removal (Lu et al., 2024; Zhang et al., 2026a) or merging (Li et al., 2024; Chen et al., 2025a).

While expert-level pruning offers deployment-friendly compression that aligns naturally with the trend toward increasingly fine-grained expert architectures, existing methods (He et al., 2025; Zhang et al., 2026a) suffer from a fundamental limitation: their uniform evaluation strategy dilutes critical signals from important tokens. Specifically, these approaches treat all input tokens equally when assessing expert importance, allowing redundant or uninformative tokens to influence pruning decisions as much as critical ones. This token-agnostic evaluation compromises the quality of expert selection, as the contributions from important tokens are obscured by noise from less relevant inputs.

Recent works on efficient inference (Chen et al., 2024; Ye et al., 2025; Zhang et al., 2026b) have shown that attention mechanisms can be used to selectively drop unimportant tokens, enabling inference acceleration with minimal im-

---

[*]Equal contribution [1]Hikvision Research Institute [2]School of Data Science and Engineering, East China Normal University, Shanghai [3]Polytechnic Institute, Zhejiang University, Hangzhou. Correspondence to: Wenming Tan <tanwenming@hikvision.com>, Jilin Hu <jlhu@dase.ecnu.edu.cn>.

*Proceedings of the $43^{rd}$ International Conference on Machine Learning*, Seoul, South Korea. PMLR 306, 2026. Copyright 2026 by the author(s).

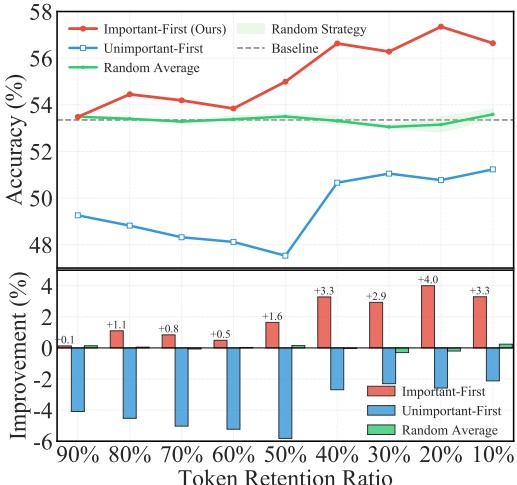

*Figure 1.* Performance of frequency-based expert pruning with different token selection strategies on OLMoE-7A1B. Guidance from important tokens (red) improves accuracy by up to 4.0% over the full-token baseline (gray dashed line), while unimportant tokens (blue) degrade performance. Random selection (green) shows negligible variation, confirming that selection quality, not quantity, determines effectiveness.

pact on model performance. Motivated by this observation, we analyze the attention patterns in OLMoE-1B-7B-0125 (abbreviated as OLMoE-7A1B) (Muennighoff et al., 2025) and observe highly sparse attention distributions: the top 20% of tokens capture approximately 80% of total attention weights (detailed analysis in Appendix H). To validate whether this sparsity translates to pruning effectiveness, we conduct frequency-based expert pruning experiments guided by tokens with different importance levels. As shown in Figure 1, pruning decisions based solely on the top 20% important tokens yield up to 4.0% accuracy improvements over full-token baselines across 8 zero-shot tasks (details in Appendix 4.1). IoU analysis (Appendix C) reveals that after token selection, the average IoU of the top-32 experts falls below 80%, with some layers dropping to 50%, indicating that token selection fundamentally reshapes pruning decisions. Additionally, our method increases expert score variance by 1.4×, reflecting a higher signal-to-noise ratio in expert scoring.This empirical evidence strongly suggests that ignoring token importance is a key factor limiting the effectiveness of existing pruning methods. This observation motivates our central question: **Can we filter out noisy tokens and identify experts that truly impact model performance based on tokens that matter most?**

To address this challenge, we propose STEP (Selective Token-guided Expert Pruning), a compression framework that evaluates expert importance based on selected tokens rather than all tokens. The key insight is that standard expert evaluation suffers from signal dilution: when aggregating importance scores across all tokens, signals from truly im-

portant tokens are overwhelmed by noise from numerous less relevant ones. Since unimportant tokens typically outnumber important ones, this noise accumulation degrades the reliability of expert evaluation. Our token selection mechanism addresses this by filtering out noise-contributing tokens before assessing expert importance.

To better evaluate the impact of these important tokens on expert importance, we perform a dual-factor scoring mechanism, which provides direct loss assessment compared to previous approaches that rely on proxy metrics (e.g., statistical metrics (He et al., 2025; Zhang et al., 2026a)). Furthermore, we convert pruned experts into bias terms instead of discarding them entirely, which helps minimize information loss while still reducing parameters. These design choices allow STEP to achieve effective compression with minimal computational overhead.

Extensive experiments across model scales (7B to 100B parameters) and four MoE architectures demonstrate the effectiveness of STEP on both language modeling and zero-shot evaluation tasks. On the 30B-parameter Qwen3 model, STEP completes the entire pruning pipeline in merely 10 minutes. At 50% pruning ratios, our approach achieves 1.5× inference speedup and nearly 50% memory reduction without requiring specialized operators or incurring noticeable performance degradation. These results establish STEP as a practical and effective solution for deploying large-scale MoE models in resource-constrained environments.

## 2. Related Work

**Mixture-of-Experts Architectures.** The scaling of large language models has led to prohibitive computational costs, making MoE architectures a practical solution by activating only a subset of experts per token. Starting from early work by Jacobs et al. (Jacobs et al., 1991) and later integration into Transformers by Shazeer et al. (Shazeer et al., 2017), modern MoE systems have achieved unprecedented scale. Recent models like DeepSeek-R1 (DeepSeek-AI, 2025) with 671B parameters, Llama 4 Maverick (Meta, 2025) activating only 17B of 400B parameters, and Qwen3-Next (Qwen Team, 2025) using 3.7% of its parameters demonstrate impressive per-token efficiency but leave most parameters inactive. However, many experts are rarely selected or functionally redundant (Lu et al., 2024; Zhang et al., 2026a), creating a critical need for efficient MoE compression techniques.

**MoE-Specific Pruning Methods.** Existing approaches fall into two categories. Matrix-level compression operates on weight matrices within experts, including adapted pruning methods like MoE-Pruner (Xie et al., 2024) and STUN (Lee et al., 2025), shared projection techniques such as Mo-LAE (Liu et al., 2025c) and MoBE (Chen et al., 2025b), and hybrid approaches including D²-MoE (Gu et al., 2025)

and ResMoE (Ai et al., 2025). Expert-level pruning directly reduces expert numbers through various strategies. Methods such as NAEE (Lu et al., 2024) and EEP (Liu et al., 2024) measure expert contributions via combinatorial search, but exhaustive enumeration becomes prohibitively expensive for modern fine-grained MoE architectures. Other approaches include one-shot expert pruning methods based on routing statistics (Koishekenov et al., 2023; He et al., 2025), dynamic expert replacement (Zhang et al., 2026a), expert merging via clustering (Li et al., 2024; Chen et al., 2025a), and adaptive grouping with joint SVD (Li et al., 2025a). While these methods effectively address parameter growth from increasing expert numbers, they evaluate expert importance uniformly across all tokens, leading to suboptimal pruning decisions where critical token signals are diluted by less relevant ones.

**Token Importance in LLMs.** Recent work has demonstrated that not all tokens contribute equally to model outputs. In the context of efficient inference, prompt compression techniques like Selective Context (Li et al., 2023) and LLMLingua (Jiang et al., 2023) use perplexity-based metrics to remove redundant tokens from input sequences. Similarly, methods like FastV (Chen et al., 2024) leverage attention weights to identify and preserve important tokens while pruning less relevant ones. Subsequent work such as Fitprune (Ye et al., 2025) and PyramidDrop (Xing et al., 2024) extends this idea with multi-stage pruning strategies. Recent studies have further revealed the disproportionate impact of certain token subsets on model capabilities. Notably, recent work (Wang et al., 2025) demonstrates that high-entropy minority tokens play a critical role in reinforcement learning for chain-of-thought reasoning. While these studies reveal token redundancy in LLMs, we are the first to explore how token importance affects expert selection and guides expert pruning in MoE-LLMs.

**Comparison with Token-Agnostic Expert Pruning.** Among expert-level pruning methods, MoNE (Zhang et al., 2026a) shares the closest problem setting with STEP, yet the two approaches diverge fundamentally in methodology and underlying philosophy. Unlike MoNE's token-agnostic strategy that uniformly aggregates signals across all calibration tokens and dilutes critical information, STEP introduces an attention-guided token selection stage to focus evaluation exclusively on the most salient tokens. This granularity shift is accompanied by a redefinition of expert importance: MoNE identifies replaceable experts via output variance and routing probability, whereas STEP directly quantifies loss-relevant importance using activation frequency and gating-weighted contribution norms. Consequently, their compensation mechanisms also differ; MoNE relies on local, closed-form empirical means per expert, while STEP employs global gradient-based optimization to learn bias vectors across all layers, ensuring coherent model-wide adjustment.

## 3. Methodology

This section introduces our method for pruning sparse Mixture-of-Experts (MoE) models. We first formalize the foundation of MoE architectures, then present a pruning framework built on a core innovation: identifying critical tokens to guide pruning decisions. To enhance this token-guided pruning strategy, we incorporate two architectural components: a dual-factor scoring mechanism for expert importance evaluation, and a bias conversion technique to mitigate performance degradation from pruned experts. The complete algorithm workflow is detailed in Algorithm 1.

**Preliminary.** MoE represent a powerful paradigm for scaling neural networks by conditionally activating subsets of parameters. In a standard MoE layer, the computation is governed by:

$$\text{MoE}(x) = \sum_{i \in \text{top-}k(G(x))} G_i(x) \cdot E_i(x) \qquad (1)$$

where $G(x) = \text{softmax}(xW_g)$ represents the gating weights determining expert activation, $E_i(x)$ denotes the $i$-th expert's computation. The gating mechanism routes each input token $x \in \mathbb{R}^d$ to the top-$k$ experts based on learned routing probabilities, where $k$ is the default configuration of MoE models.

While sparse activation improves computational efficiency, it suffers from significant memory overhead due to the linear growth of expert count $n$: each expert introduces parameter matrices that must be stored regardless of utilization.

**STEP Overview.** Our approach slims existing MoE LLMs through a framework centered on selective token guidance, as illustrated in Fig 2. While existing pruning methods treat all tokens equally when estimating expert importance, our framework addresses this limitation by employing attention-guided token selection to identify informative tokens that provide more reliable signals for pruning decisions. Building upon this token-aware foundation, we further integrate dual-factor expert scoring and knowledge-preserving expert-to-bias conversion to minimize the performance loss incurred by expert pruning.

### 3.1. Attention-Guided Token Selection

In standard MoE gating, all tokens participate in expert pruning via the router network. This causes a **signal dilution problem**: important tokens that should guide pruning are overwhelmed by numerous less relevant ones, much like informed voters being outnumbered by uninformed ones in an election. Crucially, not all tokens contribute equally to pruning decisions. Many tokens, such as function words or other semantically light tokens, introduce noise rather than signal

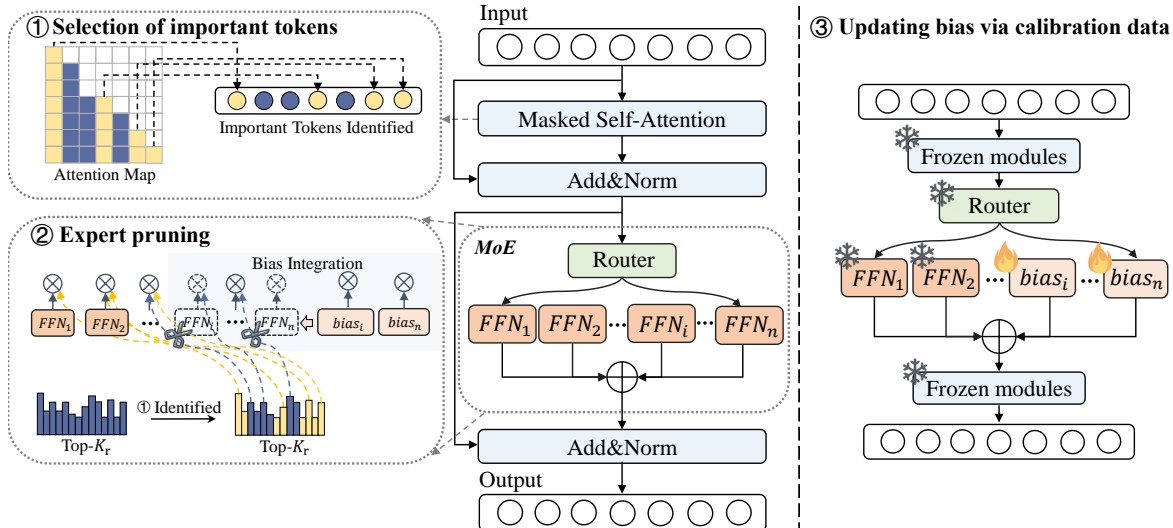

*Figure 2.* The STEP framework efficiently prunes MoE models via synergistic stages: (1) attention-guided token selection identifies important tokens for pruning; (2) dual-factor expert scoring prunes less critical experts, replacing them with trainable biases; and (3) knowledge-preserving bias updating via calibration data globally, freezing non-essential modules to maintain expressiveness.

when determining which computational pathway (expert) should be activated. This noise accumulation leads to sub-optimal routing decisions and unnecessary computational overhead.

**Theoretical Motivation.** Expert evaluation often aggregates gating scores across all tokens: $\sum_{t=1}^{T} G_i(x_t)$. We decompose this sum into signal and noise components:

$$\sum_{t=1}^{T} G_i(x_t) = \underbrace{\sum_{t \in \mathcal{T}_{\text{important}}} G_i(x_t)}_{S} + \underbrace{\sum_{t \in \mathcal{T}_{\text{unimportant}}} G_i(x_t)}_{N} \quad (2)$$

The signal-to-noise ratio is SNR $= \frac{|S|^2}{|N|^2}$. Since unimportant tokens typically outnumber important ones, $|N|$ tends to be large, leading to low SNR and unreliable expert evaluation.

Under effective token selection that keeps important tokens while discarding most unimportant ones:

$$\sum_{t \in \mathcal{T}_{\text{selected}}} G_i(x_t) = S + \epsilon \quad (3)$$

where $\epsilon$ represents minimal residual noise from any remaining unimportant tokens. The selected SNR becomes:

$$\text{SNR}_{\text{selected}} = \frac{|S|^2}{|\epsilon|^2} \quad (4)$$

Since $|\epsilon| < |N|$ due to noise reduction, we obtain $\text{SNR}_{\text{selected}} > \text{SNR}_{\text{original}}$. This improvement in signal-to-noise ratio leads to more reliable expert evaluation.

**Implementation.** For layer $\ell$ with $T$ input tokens, we score token importance using attention weights. While per-token

gradients directly measure loss contributions, computing them requires full backward propagation through the entire network, which is computationally expensive for LLMs. Attention weights offer an efficient forward-only alternative supported by prior work (He et al., 2024; Chen et al., 2024; Xing et al., 2024). In Appendix D, we rigorously justify this choice through gradient flow analysis and ablation studies on different token selection metrics.

Following prior work (Chen et al., 2024; Xing et al., 2024), we derive token importance scores $\alpha^{(\ell)}$ from the attention distribution and form the selected token set by choosing the top-$\lceil \tau \cdot T \rceil$ tokens:

$$\mathcal{T}_{\text{selected}}^{(\ell)} = \left\{ x_t \mid \alpha_t^{(\ell)} \in \text{top-}\lceil \tau \cdot T \rceil \text{ of } \alpha^{(\ell)} \right\} \quad (5)$$

where $\tau$ is the selection ratio. Expert importance scoring operates only on $\mathcal{T}$selected, concentrating evaluation on contextually relevant tokens while filtering noise from less informative ones. This rebalances voting weights from uniform $\frac{1}{T}$ to concentrated $\frac{1}{|\mathcal{T}_{\text{selected}}|}$, where $|\mathcal{T}_{\text{selected}}| < T$.

### 3.2. Expert importance scoring

Recent MoE expert pruning relies on statistical metrics (Koishekenov et al., 2023; Zhuang et al., 2024; Zhang et al., 2026a) that fail to capture the loss impact of experts in MoE architectures. Our dual-factor scoring addresses this by incorporating both expert activation frequency and feature contribution, providing a more nuanced understanding of each expert's role in the model's computational graph. An expert that is rarely activated but produces high-magnitude

outputs may be crucial for handling rare yet important cases, whereas a frequently activated expert with consistently low impact may be redundant.

Given an input token $x$, the MoE layer computation explicitly reveals these two critical dimensions:

$$\text{MoE}^{(\ell)}(x) = \sum_{\underbrace{i \in \text{top-}k\left(G^{(\ell)}(x)\right)}_{\textbf{Frequency}}} \underbrace{G_i^{(\ell)}(x) \cdot E_i^{(\ell)}(x)}_{\textbf{Feature}} \quad (6)$$

To evaluate experts within $\mathcal{T}_{\text{selected}}$, we propose a dual-factor metric that combines activation frequency and feature significance, both of which impact the MoE-layer loss:

$$F_i^{(\ell)} = \frac{1}{|\mathcal{T}_{\text{selected}}^{(\ell)}|} \sum_t \mathbf{1}\left[G_i^{(\ell)}(x_t) \in \text{top-}k\left(G^{(\ell)}(x_t)\right)\right] \quad (7)$$

$$L_i^{(\ell)} = \frac{1}{|\mathcal{T}_{\text{selected}}^{(\ell)}|} \sum_t \left\| G_i^{(\ell)}(x_t) E_i^{(\ell)}(x_t) \right\|_2 \quad (8)$$

$$\text{Score}(E_i^{(\ell)}) = F_i^{(\ell)} \cdot L_i^{(\ell)} \quad (9)$$

The activation frequency $F_i^{(\ell)}$ measures how often expert $i$ appears among the top-$k$ experts, reflecting its overall utility. The feature norm $L_i^{(\ell)}$ quantifies the magnitude of its contribution to the layer output, weighted by gating probability. Combining these metrics ensures retention of experts that are both consistently useful and impactful. Within each layer, experts are ranked by this metric, and the lowest-scoring ones are removed according to pruning ratio $p$.

### 3.3. Expert-to-Bias Knowledge Preservation

Since the full computational complexity of an expert is often unnecessary (Lu et al., 2024; Zeng et al., 2024; Jin et al., 2025) and its core contribution for output can be preserved in a lightweight vector (Zhang et al., 2026a), we mitigate irreversible information loss from pruning by transforming essential computational patterns of removed experts into compact bias representations. This plug-and-play bias ($<$ 0.1% of original weights) mitigates the performance degradation caused by expert pruning without costly updates.

**Bias Vector Initialization and Integration.** For each pruned expert $E_i^{(\ell)} \in \mathcal{R}^{(\ell)}$, we convert it into a bias vector that captures its typical output pattern. The bias vector is computed by averaging the expert's outputs on tokens where it is highly activated:

$$\mathbf{b}_i^{(\ell)} = \frac{1}{|\mathcal{T}_i^{(\ell)}|} \sum_{x_t \in \mathcal{T}_i^{(\ell)}} E_i^{(\ell)}(x_t),$$
$$\mathcal{T}_i^{(\ell)} = \{x_t \mid G_i^{(\ell)}(x_t) \in \text{top-}k(G^{(\ell)}(x_t))\} \quad (10)$$

where $\mathcal{T}_i^{(\ell)} \subseteq \mathcal{T}_{selected}^{(\ell)}$ contains the selected tokens that most strongly activate expert $i$. This formulation has a clear interpretation: we solve for the optimal constant approximation under squared loss. Specifically, $\mathbf{b}_i^{(\ell)}$ minimizes $\sum_m ||E_i(x_m) - b||_2^2$ over the calibration set, yielding the closed-form solution above.

After converting pruned experts to bias vectors, the layer output becomes:

$$\tilde{E}_j^{(\ell)}(x) = \begin{cases} E_j^{(\ell)}(x), & E_j^{(\ell)} \notin \mathcal{R}^{(\ell)} \\ \mathbf{b}_j^{(\ell)}, & E_j^{(\ell)} \in \mathcal{R}^{(\ell)} \end{cases} \quad (11)$$

$$\text{Output}^{(\ell)}(x) = \sum_{j \in \text{top-}k(G(x))} G_j^{(\ell)}(x) \cdot \tilde{E}_j^{(\ell)}(x) \quad (12)$$

This formulation maintains the original gating structure while replacing pruned experts with their bias vectors.

**Bias Vector Refinement.** While the initialization above provides layer-wise optimal bias vectors, we refine them through a brief fine-tuning phase to achieve a global optimum across the entire model. This refinement is computationally efficient because the bias vectors start from near-optimal initial values obtained during layer-wise optimization, and the pruned experts have minimal impact on loss due to their low Freq $\times$ ||Output|| scores. Consequently, only minor adjustments on a small calibration dataset are needed for convergence as detailed in Appendix F.4. We acknowledge that while more extensive fine-tuning with larger datasets could potentially yield further improvements, our lightweight approach offers a practical trade-off between performance and computational cost. We optimize the bias vectors while keeping all expert parameters frozen:

$$\min_b \sum_{(x,y) \in \mathcal{D}_{\text{cal}}} \mathcal{L}(f_b'(x), y) \quad (13)$$

where $b$ denotes the bias vectors added to compensate for the removed experts. The remaining expert parameters are kept frozen during this optimization.

## 4. Experiments

In this section, we conduct comprehensive experiments to evaluate STEP. We first compare our approach against state-of-the-art pruning baselines across multiple MoE architectures and sparsity ratios. We then perform ablation studies to analyze the contribution of each component, and demonstrate the practical efficiency gains in terms of resource consumption and inference speedup. All methods compared in this paper follow the same static offline pruning paradigm: expert weights are permanently removed post-calibration,

and inference is performed on a fixed reduced architecture with zero runtime overhead.

## 4.1. Experimental Setup

We evaluate STEP on four MoE architectures: OLMoE-7A1B (Muennighoff et al., 2025), Moonlight-16A3B (Liu et al., 2025a), Qwen3-30A3B (Yang et al., 2025), and GLM-Air (Team et al., 2025), ranging from 7B to 106B total parameters with activation ratios from 1B to 12B. Performance is assessed via perplexity on C4 (Raffel et al., 2020) and WikiText (Merity et al., 2016), zero-shot reasoning on eight benchmarks including PIQA (Bisk et al., 2020), BoolQ (Clark et al., 2019), HellaSwag (Zellers et al., 2019), WinoGrande (Sakaguchi et al., 2021), ARC-Easy and ARC-Challenge (Clark et al., 2018), OpenBookQA (Mihaylov et al., 2018), and MMLU (Hendrycks et al., 2021), and specialized tasks with HumanEval (Chen et al., 2021) and GSM8K (Cobbe et al., 2021). We compare against representative baselines: $D^2$-MoE (Gu et al., 2025), MoNE (Zhang et al., 2026a), Gate Score (He et al., 2025), MC-SMoE (Li et al., 2024), and HC-SMoE (Chen et al., 2025a). Complete experimental details are in Appendix B.

## 4.2. Pruning Performance and Comparisons

**Zero-Shot Task Performance.** Our method demonstrates consistent superiority across zero-shot task generalization with notable gains in average accuracy over existing approaches as detailed in Table 1. On OLMoE-7A1B at 25% sparsity, we achieve an average accuracy of 58.42%, outperforming HC-SMoE by 3.54% and MoNE by 5.16%. This improvement is reflected across diverse benchmarks where our approach either leads or remains competitive, consistently ranking first or second on individual datasets. Furthermore, scalability is evident in experiments conducted on Qwen3-30A3B with 64 experts, where our method attains an average accuracy of 65.78%, exceeding MoNE by 1.63% even under 50% expert reduction. Experiments on a 100B-parameter model (Appendix E) further validate scalability to larger architectures.

**Scaling Analysis of Expert Pruning.** As shown in Figure 3, our method maintains consistent advantages over baselines when reducing experts from 64 to 20. The performance gap widens progressively with compression ratio, achieving a maximum 5.7% improvement in average accuracy. Our approach shows linear perplexity growth while competitors degrade exponentially (Appendix F.1), confirming the robustness of our strategy. STEP consistently outperforms baselines across all scales and sparsity levels, achieving up to 27% relative perplexity reduction versus MoNE on OLMoE-7A1B at 50% sparsity. Notably, at aggressive 70% sparsity, our method achieves a remarkable 72% perplexity reduction. In contrast, merging-based methods (MC-SMoE,

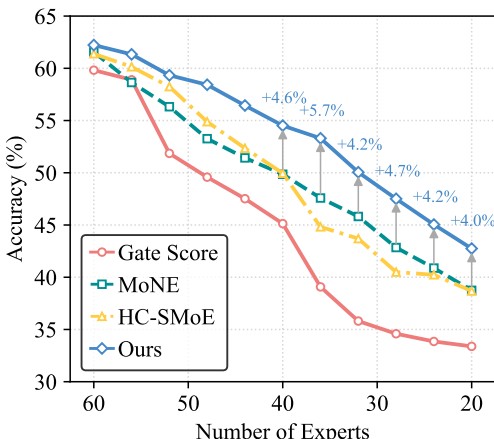

*Figure 3.* Average accuracy of OLMoE-7A1B across 8 zero-shot task benchmarks under different numbers of remaining experts and pruning methods.

HC-SMoE) show substantial degradation, demonstrating that preserving experts is superior to merging them. This advantage increases at higher pruning ratios.

**Code and Math Task Evaluation.** We evaluate STEP on code generation (HumanEval) and mathematical reasoning (GSM8K). Results are shown in Table 2. STEP retains 98.9% of dense model performance, outperforming all baselines. This shows our bias vectors preserve expert knowledge effectively for out-of-distribution tasks. At 50% sparsity, STEP achieves 59.61% average accuracy, exceeding MoNE by 7.2 points and HC-SMoE by 15.1 points. Although generation tasks exhibit greater degradation than classification, STEP maintains the best retention.

## 4.3. Ablation Analysis

**Component Contribution.** As detailed in Table 3, incorporating **token selection** yields +1.64% average gain versus baseline. Combining **dual-factor metric** with **expert-to-bias** achieves +2.58% improvement, while adding token importance to this configuration provides an additional +2.26% boost. Full integration delivers optimal performance, demonstrating component synergy.

**Calibration Robustness.** We further assess method stability across different data scales. All evaluated methods maintain perplexity variance below 0.3 and ARC-C accuracy variance under 1% across sample sizes from 32 to 512 sequences, except for MC-SMoE, which exhibits substantially higher variances of 7.4 and 5.1%, respectively, as shown in Figure 4. This comparative analysis confirms our method's insensitivity to calibration data scale, thereby enhancing practical deployability.

**Expert Activation Frequency Distribution.** Figure 5 visualizes the routing distributions across layers before and after

*Table 1.* Zero-Shot Task Accuracy(%) Comparison Across Models Under Structured Pruning at 25% and 50% MoE Pruning Ratios. m experts with n% density denoted as $m_{n\%}$, Bold indicates best performance, underlined indicates second-best.

| Model | Experts | Method | ARC-E | ARC-C | BoolQ | PIQA | Wino | Hella | MMLU | OBQA | Avg |
|---|---|---|---|---|---|---|---|---|---|---|---|
| **OLMoE-7A1B** | 64 | Original | 77.23 | 46.84 | 70.15 | 78.62 | 69.06 | 58.44 | 53.54 | 44.20 | 62.26 |
| | $64_{75\%}$ | $D^2$-MoE(ICML25) | 71.21 | 37.46 | 66.67 | 75.35 | 66.85 | 49.63 | 28.39 | 41.20 | 54.60 |
| | | MC-SMoE(ICLR24) | 70.12 | 38.91 | 52.26 | 66.32 | 61.96 | 41.47 | 37.83 | 35.90 | 50.60 |
| | | HC-SMoE(ICML25) | 71.25 | 39.59 | 65.78 | 72.42 | 66.22 | 51.69 | **42.65** | 29.40 | 54.88 |
| | 48 | Frequency | 62.37 | 34.04 | 62.94 | 75.89 | **67.32** | 53.11 | 31.13 | 41.00 | 53.48 |
| | | Gate Score(TMLR25) | 54.88 | 28.58 | 62.11 | 73.12 | 63.30 | 50.22 | 24.50 | 39.90 | 49.58 |
| | | MoNE(ICLR26) | 62.37 | 33.28 | 65.84 | 77.09 | 67.32 | 54.88 | 23.27 | 42.00 | 53.26 |
| | | Ours | **73.82** | **44.03** | **68.35** | **77.53** | 65.51 | **55.24** | 39.70 | **43.20** | **58.42** |
| | $64_{50\%}$ | $D^2$-MoE(ICML25) | 60.37 | 29.36 | 62.03 | 69.64 | 58.93 | 39.70 | 25.81 | 29.20 | 46.88 |
| | | MC-SMoE(ICLR24) | 30.81 | 18.52 | 42.66 | 55.60 | 50.36 | 27.41 | 23.00 | 26.90 | 34.41 |
| | | HC-SMoE(ICML25) | 46.84 | 26.28 | 58.20 | 62.19 | 59.12 | 39.08 | **29.60** | 28.20 | 43.69 |
| | 32 | Frequency | 36.91 | 22.27 | 43.40 | 62.51 | 51.70 | 36.12 | 22.90 | 27.20 | 37.89 |
| | | Gate Score(TMLR25) | 34.93 | 19.03 | 43.85 | 59.47 | 50.83 | 31.03 | 23.55 | 23.70 | 35.80 |
| | | MoNE(ICLR26) | 48.91 | 24.74 | 62.39 | 71.11 | **59.67** | 44.96 | 23.10 | 31.60 | 45.81 |
| | | Ours | **64.10** | **31.40** | **63.52** | **73.88** | 58.72 | **46.17** | 26.40 | **36.20** | **50.05** |
| **Moonlight-16A3B** | 64 | Original | 84.51 | 56.14 | 80.37 | 78.94 | 71.11 | 59.27 | 67.29 | 45.00 | 67.83 |
| | $64_{75\%}$ | $D^2$-MoE(ICML25) | 78.20 | 45.22 | 74.80 | 76.55 | 69.38 | 49.89 | **53.74** | 42.20 | 61.25 |
| | | MC-SMoE(ICLR24) | 77.36 | 45.22 | 79.08 | 78.40 | **72.14** | 56.15 | 49.00 | 33.60 | 61.37 |
| | | HC-SMoE(ICML25) | 69.32 | 37.20 | 69.11 | 68.99 | 59.12 | 46.00 | 52.55 | 34.80 | 53.89 |
| | 48 | Frequency | 81.39 | 51.19 | 77.86 | 79.49 | 71.19 | 58.72 | 48.68 | 44.40 | 64.12 |
| | | Gate Score(TMLR25) | 78.45 | 48.12 | 77.58 | 79.71 | 70.96 | 58.52 | 50.66 | 45.20 | 63.65 |
| | | MoNE(ICLR26) | 80.89 | 50.94 | 78.53 | 80.03 | 70.72 | **58.88** | 49.25 | 45.40 | 64.33 |
| | | Ours | **82.17** | **52.29** | **79.17** | **80.10** | 71.03 | **58.88** | 52.62 | **45.90** | **65.27** |
| | $64_{50\%}$ | $D^2$-MoE(ICML25) | 65.57 | 33.96 | 67.74 | 69.80 | 61.96 | 41.25 | **32.92** | 33.80 | 50.88 |
| | | MC-SMoE(ICLR24) | 62.58 | 28.24 | 63.24 | 68.33 | 59.51 | 38.97 | 22.90 | 27.20 | 46.37 |
| | | HC-SMoE(ICML25) | 53.54 | 24.74 | 60.89 | 62.30 | 50.99 | 30.68 | 30.81 | 30.00 | 42.99 |
| | 32 | Frequency | 61.78 | 30.20 | 59.08 | 72.42 | 63.38 | 46.56 | 22.88 | 38.00 | 49.28 |
| | | Gate Score(TMLR25) | 62.37 | 31.57 | 54.68 | 72.80 | 63.30 | 46.17 | 22.92 | 36.20 | 48.75 |
| | | MoNE(ICLR26) | 66.75 | 33.79 | 71.13 | 77.48 | **70.56** | 53.06 | 22.97 | **40.00** | 54.47 |
| | | Ours | **71.14** | **38.48** | **74.09** | **77.64** | 68.14 | **54.40** | 25.87 | 39.80 | **56.20** |
| **Qwen3-30A3B** | 128 | Original | 79.84 | 53.24 | 88.62 | 79.38 | 70.64 | 59.51 | 77.79 | 44.60 | 69.20 |
| | $128_{75\%}$ | $D^2$-MoE(ICML25) | **80.56** | 54.01 | 88.56 | 79.43 | 70.17 | 59.61 | **73.78** | 44.00 | 68.77 |
| | | MC-SMoE(ICLR24) | 64.69 | 39.16 | 77.65 | 67.08 | 58.80 | 41.45 | 54.00 | 31.80 | 54.33 |
| | | HC-SMoE(ICML25) | 75.97 | 45.82 | 86.54 | 76.22 | 70.40 | 51.54 | 66.89 | 41.00 | 64.30 |
| | 96 | Frequency | 80.36 | 54.75 | 88.36 | 79.67 | 70.20 | 59.25 | 73.31 | 44.00 | 68.74 |
| | | Gate Score(TMLR25) | 77.99 | 50.77 | 88.50 | 79.43 | 69.93 | 59.05 | 72.77 | 42.60 | 67.63 |
| | | MoNE (ICLR26) | 80.35 | 54.27 | **89.11** | 79.87 | 70.48 | 59.52 | 72.83 | 44.20 | 68.83 |
| | | Ours | 80.43 | 54.69 | 88.83 | **79.98** | 70.80 | **59.84** | 72.90 | **45.00** | **69.06** |
| | $128_{50\%}$ | $D^2$-MoE(ICML25) | 66.12 | 40.19 | 79.51 | 75.03 | 64.09 | 53.37 | 34.06 | 38.00 | 56.30 |
| | | MC-SMoE(ICLR24) | 30.89 | 19.45 | 58.32 | 55.11 | 46.96 | 27.32 | 26.08 | 25.00 | 36.14 |
| | | HC-SMoE(ICML25) | 65.19 | 35.84 | 83.03 | 70.95 | 65.19 | 41.84 | 46.06 | 36.20 | 55.54 |
| | 64 | Frequency | 42.26 | 22.95 | 65.45 | 77.74 | 63.69 | 52.96 | 48.98 | 37.20 | 51.40 |
| | | Gate Score(TMLR25) | 48.40 | 28.07 | 82.23 | 69.64 | 61.88 | 48.29 | 42.00 | 32.80 | 51.66 |
| | | MoNE(ICLR26) | 76.56 | 46.76 | 87.61 | 79.54 | 69.85 | 57.97 | 51.67 | 43.20 | 64.15 |
| | | Ours | **78.58** | **47.70** | **88.04** | **79.61** | **70.33** | **58.24** | **58.72** | **45.00** | **65.78** |

expert pruning for Moonlight-16A3B on the C4, Wikitext2, and Arc-Challenge datasets at 25% pruning ratios. The expert activation frequencies exhibit minimal divergence between pre-pruning and post-pruning states, indicating that our pruning approach does not destabilize the routing mechanism. Additional visualizations across different datasets, models and sparsity in Appendix I confirm this routing stability.

**Hyperparameter Sensitivity.** Additional ablations on **token selection ratio**, **experts-to-bias effectiveness**, and **training epochs** in Appendix F confirm robustness across hyperparameter variations and guide our settings.

### 4.4. Resource Consumption and Speedup

To assess the practical efficiency of our pruning algorithm, we evaluate computational overhead, memory usage, and inference performance across multiple architectures and pruning ratios, using a consistent token configuration (1024 for prefill, 256 for decoding). Empirical results from Table 4 and Figure 6 on Qwen3-30A3B show significant improvements: our method reduces memory footprint and parameter count by 23.4%–47.5%, while achieving 1.10–1.50× inference speedup in both prefill and decoding stages. Additionally, the entire process is completed within 10 minutes with a negligible peak memory increase (e.g., 59.84GB vs. 57.09GB), incurring minimal resource overhead. The linear

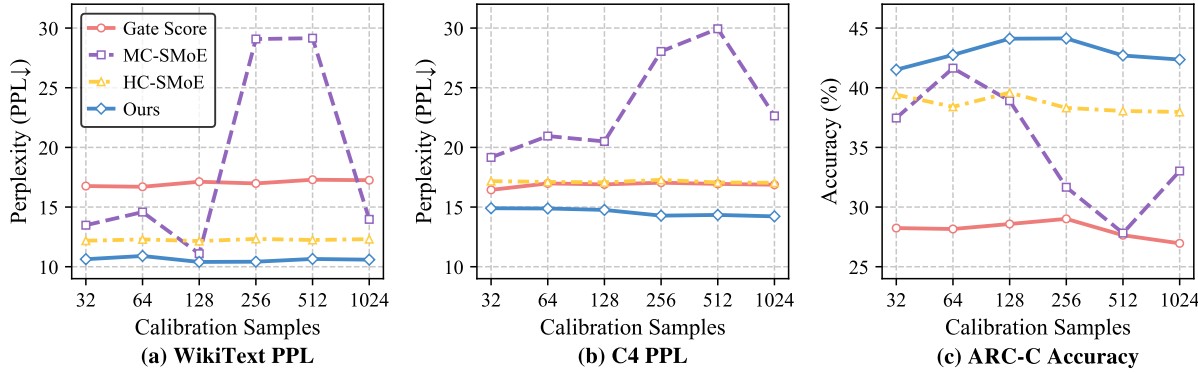

*Figure 4.* Performance of OLMoE-7A1B under varying numbers of calibration samples and different pruning methods.

*Table 2.* Performance on generation tasks using HumanEval pass@1 and GSM8K accuracy.

| Model | Experts | Method | HumanEval | GSM8K | Avg |
|---|---|---|---|---|---|
| | 128 | – | 94.12 | 89.92 | 92.02 |
| **Qwen3-30A3B** | 96 | Frequency | 92.47 | 88.20 | 90.34 |
| | | HC-SMoE | 92.48 | 87.49 | 89.99 |
| | | MoNE | 93.09 | 88.17 | 90.63 |
| | | **STEP** | **93.42** | **88.63** | **91.03** |
| | 64 | Frequency | 59.35 | 44.20 | 51.78 |
| | | HC-SMoE | 34.15 | 54.97 | 44.56 |
| | | MoNE | 63.78 | 41.09 | 52.44 |
| | | **STEP** | **64.02** | **55.19** | **59.61** |

*Table 3.* Ablation study on three core components with 25% experts pruned on OLMoE-7A1B, and the hyper-parameter $\tau$ is set to 0.5. Note: ✓ indicates inclusion and ✗ indicates exclusion.

| Method Components | | | 8 Zero-shot |
|---|---|---|---|
| Token Selection | Dual-factor | Experts-to-Bias | Task Average |
| ✗ | ✗ | ✗ | 53.48 |
| ✓ | ✗ | ✗ | 55.12 |
| ✗ | ✓ | ✗ | 55.43 |
| ✗ | ✓ | ✓ | 56.16 |
| ✓ | ✓ | ✗ | 57.51 |
| ✓ | ✓ | ✓ | **58.42** |

correlation between pruning ratio and resource reduction underscores the predictability and deployment suitability of our approach under constrained resource budgets. Further evaluation on OLMoE-7A1B model, detailed in Appendix G, confirms the generalizability of efficiency gains.

## 5. Conclusion

As Mixture-of-Experts (MoE) architectures represent an emerging trend in scaling LLMs with superior performance at reduced computational cost, their substantial memory requirements pose significant challenges to practical deployment. In this paper, we propose STEP, a token-guided MoE compression method that pioneers token importance-

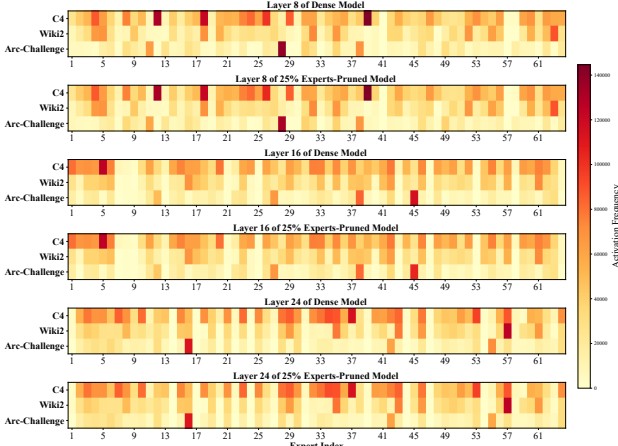

*Figure 5.* Expert activation frequencies across layers before and after expert pruning for Moonlight-16A3B on the C4, Wikitext2, and Arc-Challenge datasets.

based pruning for MoE models, with theoretical analysis validating its effectiveness. Comprehensive experiments demonstrate that STEP significantly outperforms existing MoE pruning approaches across various metrics and model scales. Notably, applying STEP to the Qwen3-30A3B model achieves nearly 50% memory compression and a 1.5× speedup while incurring only minimal performance degradation, completing the entire pruning process within 10 minutes. These outcomes enable low-cost and economical deployment of MoE models in practical applications.

**Limitations**: Our key contribution is demonstrating how token importance influences expert pruning decisions. Using established attention-based metrics, we show that token-aware pruning consistently surpasses token-agnostic approaches. While performance gains are marginal when pruning few experts (e.g., +0.23% over the best baseline with 48 experts pruned in Qwen3-30A3B), our method maintains clear advantages at higher pruning rates (e.g., 32 experts), proving its practical value in compression-heavy deployment scenarios.

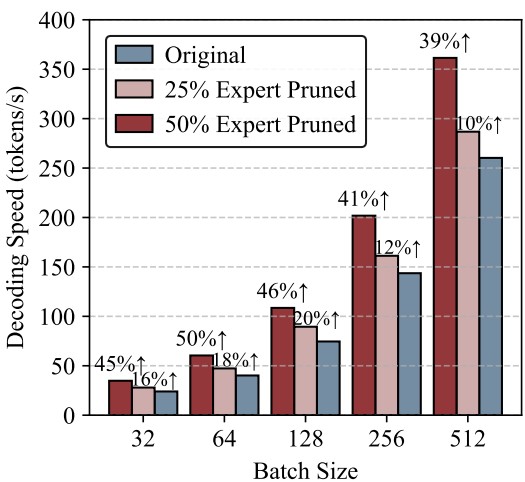

*Figure 6.* Throughput of Qwen3-30A3B under different expert pruning ratios and different batch sizes.

*Table 4.* Prefill Performance and Resource Consumption Analysis of Qwen3-30A3B.

| Metric/Experts | 128 Experts | 96 Experts | 64 Experts |
|---|---|---|---|
| **Resource Consumption** | | | |
| Memory (GB) | 57.09 | 43.46 ($\downarrow$ 24%) | 29.96 ($\downarrow$ 48%) |
| Parameters (B) | 30.53 | 23.29 ($\downarrow$ 24%) | 16.04 ($\downarrow$ 48%) |
| **Prefill Performance** | | | |
| Latency (ms) | 183.14 | 164.64 | 139.49 |
| Speedup | 1.00$\times$ | 1.11$\times$ | 1.31$\times$ |
| **Pruning Overhead** | | | |
| Prune Time (min) | – | | 3 |
| Bias Time (min) | – | | 7 |
| Memory (GB) | – | | 59.84 |

## Impact Statement

This work develops token-aware expert pruning methods to improve the efficiency of Mixture-of-Experts models. Beyond computational benefits, we hope this work encourages researchers to consider token-level dynamics when designing compression techniques for MoE architectures. Our approach reduces computational costs and memory footprint, making large language models more accessible for resource-constrained applications and supporting environmentally sustainable AI deployment. All experiments use publicly available benchmarks and datasets to ensure reproducibility. As a model compression technique, this work does not introduce new societal risks beyond those inherent to large language models themselves.

## Acknowledgments

This work was supported in part by the National Natural Science Foundation of China (Grant No. 62472174) and by the ECNU Multifunctional Platform for Innovation (Grant No. 001).

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

# Appendix

This appendix provides supporting materials for the main paper. Section A presents the complete algorithm pseudocode. Section 4.1 details experimental configurations and baseline implementations. Section D offers theoretical justification for using attention weights as token importance measures. Section E reports results on 100B+ parameter models. Section F presents additional ablation studies on key hyperparameters. Section G analyzes inference performance and resource consumption for OLMoE-7B-A1B. Section H visualizes attention patterns across layers, and Section I visualizes routing distributions across layers before and after expert pruning.

# A. Algorithm Pseudocode

We provide the complete algorithm for STEP in Algorithm 1, which outlines the three-stage pruning process described in Section 3

---

**Algorithm 1** STEP: Selective Token-guided Expert Pruning for MoE LLMs

---

**Require:** MoE model $f$, calibration dataset $\mathcal{D}_{\text{cal}}$, pruning ratio $p$, token retain threshold $\tau$
**Ensure:** Pruned $f'$ with preserved performance

1: **for** layer $\ell = 1$ to $L$ **do**

2: $\quad \mathcal{T}_{selected}^{(\ell)} \leftarrow \left\{ x_t \mid t \in \arg \text{topk}_{\lceil \tau \cdot T \rceil} \left( \alpha^{(\ell)} \right) \right\}$ {**Stage 1. Token Selection**}

3: $\quad$ **for** each expert $E_i^{(\ell)}$ **do**

4: $\quad\quad F_i^{(\ell)} \leftarrow \frac{1}{|\mathcal{T}_{selected}^{(\ell)}|} \sum_t \mathbf{1}[G_i^{(\ell)}(x_t) \in \text{top-}k \left( G^{(\ell)}(x_t) \right)]$

5: $\quad\quad L_i^{(\ell)} \leftarrow \frac{1}{|\mathcal{T}_{selected}^{(\ell)}|} \sum_t \|G_i^{(\ell)}(x_t) \cdot E_i^{(\ell)}(x_t)\|_2^2$

6: $\quad\quad \text{Score}_i^{(\ell)} \leftarrow F_i^{(\ell)} \cdot L_i^{(\ell)}$

7: $\quad$ **end for**

8: $\quad \mathcal{R}^{(\ell)} \leftarrow \{E_i^{(\ell)} \mid \text{Score}_i^{(\ell)} \in \text{lowest } p\%\}$ {**Stage 2. Expert Importance Scoring**}

9: $\quad$ **for** $E_i^{(\ell)} \in \mathcal{R}^{(\ell)}$ **do**

10: $\quad\quad \mathbf{b}_i^{(\ell)} \leftarrow \frac{1}{|\mathcal{T}_i^{(\ell)}|} \sum_{x_t} E_i^{(\ell)}(x_t)$ where $\mathcal{T}_i^{(\ell)} = \{x_t \mid G_i^{(\ell)}(x_t) \in \text{top-}k((G^{(\ell)}(x_t)))\}$

11: $\quad$ **end for**

12: $\quad$ Remove $\mathcal{R}^{(\ell)}$ from $f^{(\ell)}$

13: **end for**

14: Modify $f'$: $\text{Output}^{(\ell)} \leftarrow \sum_j G_j^{(\ell)}(x) \cdot \left( \mathbf{1}[E_j^{(\ell)} \notin \mathcal{R}^{(\ell)}] E_j^{(\ell)}(x) + \mathbf{1}[E_j^{(\ell)} \in \mathcal{R}^{(\ell)}] \mathbf{b}_j^{(\ell)} \right)$

15: $\Theta^* \leftarrow \arg \min_{\Theta} \sum_{(x,y) \in \mathcal{D}_{\text{cal}}} \mathcal{L}(f'_{\Theta}(x), y)$ {**Stage 3. Expert to Bias Converting**}

$\quad$ **return** $f'_{\Theta^*}$

---

# B. Experimental Setup and Baselines

### B.1. Experimental Setup

For fair comparison, all methods share the same calibration data: 128 randomly sampled sequences (2,048 tokens each) from C4 (Raffel et al., 2020) and CodeAlpaca (Chaudhary, 2023) for specialized domains, following established pruning practices (Sun et al., 2024; Lu et al., 2024; Lasby et al., 2025).

Three representative MoE architectures spanning different scales were evaluated: the compact OLMoE-7A1B[1] (7B parameters, activates 1B parameters, 64 experts) (Muennighoff et al., 2025), medium-scale Moonlight-16A3B[2] (16B parameters, activates 3B parameters, 64 experts) (Liu et al., 2025a), and large-scale Qwen3-30A3B[3] (30B parameters, activates 3B parameters, 128 experts) (Yang et al., 2025). All models were implemented via Hugging Face Transformers (Wolf et al., 2020) using official configurations, and all the experiments could run on a single A100 GPU.

---

[1]https://huggingface.co/allenai/OLMoE-1B-7B-0125
[2]https://huggingface.co/moonshotai/Moonlight-16B-A3B
[3]https://huggingface.co/Qwen/Qwen3-30B-A3B

We evaluate model performance across three dimensions. For intrinsic language modeling, we measure perplexity on C4 (Raffel et al., 2020) and WikiText (Merity et al., 2016). For zero-shot reasoning, we test on eight benchmarks: PIQA (Bisk et al., 2020), BoolQ (Clark et al., 2019), HellaSwag (Zellers et al., 2019), WinoGrande (Sakaguchi et al., 2021), ARC-Easy and ARC-Challenge (Clark et al., 2018), OpenBookQA (Mihaylov et al., 2018), and MMLU (Hendrycks et al., 2021). For code generation and mathematical reasoning, we use HumanEval (Chen et al., 2021) pass@1 and GSM8K (Cobbe et al., 2021) with 5-shot prompting. All reasoning tasks are evaluated using the lm-evaluation-harness framework (Gao et al., 2024).

### B.2. Baseline Comparisons

STEP was rigorously compared against state-of-the-art MoE compression approaches across three methodological categories: (1) **Intra-expert compression** represented by $D^2$-MoE (Gu et al., 2025) (delta decompression only, excluding its structured pruning on merged base weights for fair comparison); (2) **Expert pruning** including Gate Score based pruning (He et al., 2025) and state-of-the-art MoNE (Zhang et al., 2026a) which converts redundant experts into lightweight novices; (3) **Expert merging** featuring MC-SMoE (Li et al., 2024) (routing-guided expert merging with decomposition) and HC-SMoE (Chen et al., 2025a) (hierarchical feature clustering). All baselines were evaluated under identical sparsity configurations.

## C. Empirical Evidence for Token Selection Impact

We conducted Intersection over Union (IoU) analysis comparing expert rankings between token-aware and token-agnostic methods across all layers in Figure 7. After applying token selection, the average IoU of the remaining top-32 experts falls below the 80% threshold, with certain layers exhibiting IoU values as low as 50%, demonstrating that critical token selection fundamentally reshapes pruning decisions. More importantly, our method increases the variance of expert score distributions by 1.4× on average, thereby enhancing the SNR of expert scoring.

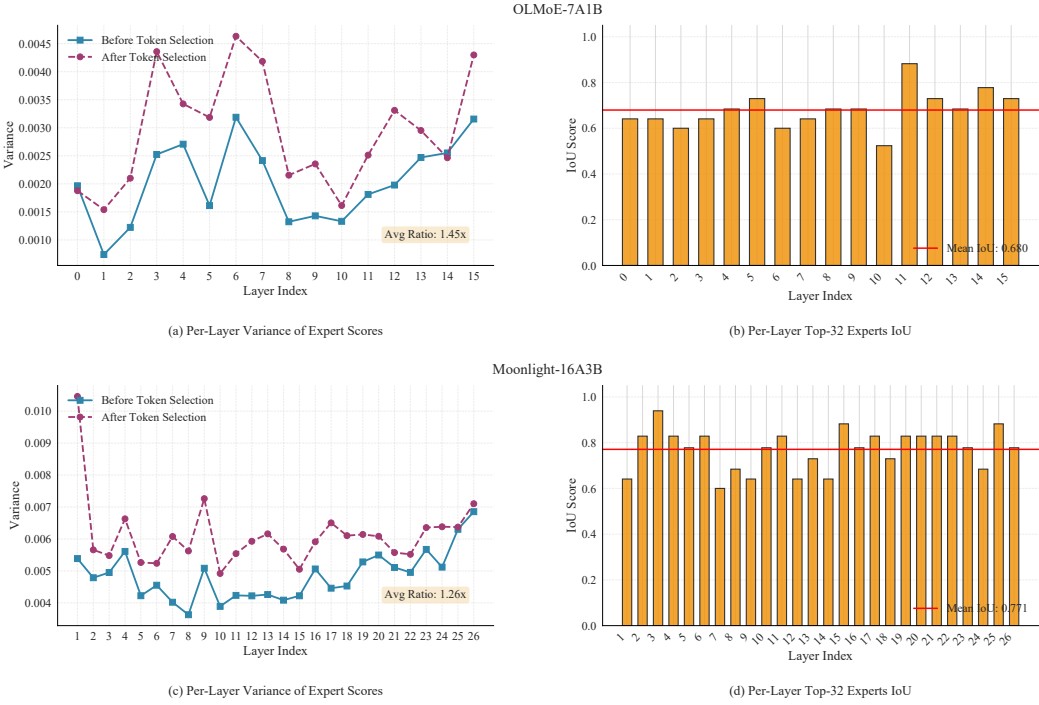

*Figure 7.* Empirical Assessment on Expert Pruning: Variance & Pruning Outcome Changes. (a) and (c) show the per-layer expert score variance for OLMoE-7A1B and Moonlight-16A3B, respectively. (b) and (d) illustrate the IoU of the top-32 experts (retained after pruning). IoU is computed as the ratio of experts consistently selected before and after pruning to the total number of retained 32 experts.

# D. Justification for Attention-Based Token Selection

### D.1. Theoretical Motivation

We show that gradient descent naturally assigns high attention weights to tokens that consistently help reduce loss. Consider a Transformer attention head with input sequence $X = [x_1, ..., x_t]$ and learnable parameters $W^Q, W^K, W^V$. The attention score is:

$$s_{ij} = \frac{(x_i W^Q)(x_j W^K)^T}{\sqrt{d_k}}$$

After softmax normalization:

$$a_{ij} = \frac{e^{s_{ij}}}{\sum_k e^{s_{ik}}}$$

During backpropagation, the gradient of loss $L$ with respect to $s_{ij}$ is:

$$\frac{\partial L}{\partial s_{ij}} = a_{ij} \left( \frac{\partial L}{\partial a_{ij}} - \sum_k a_{ik} \frac{\partial L}{\partial a_{ik}} \right)$$

When token $j$ consistently reduces loss, meaning $\frac{\partial L}{\partial a_{ij}}$ is negative with large magnitude, the gradient $\frac{\partial L}{\partial s_{ij}}$ becomes negative. Under gradient descent:

$$W^Q \leftarrow W^Q - \eta \frac{\partial L}{\partial W^Q}, \text{ where } \frac{\partial L}{\partial W^Q} \propto \sum_{ij} \frac{\partial L}{\partial s_{ij}} \cdot \frac{\partial s_{ij}}{\partial W^Q}$$

Since $s_{ij}$ increases with $W^Q$ and $W^K$, negative gradients cause $s_{ij}$ to grow during training. The softmax normalization creates competition among tokens, allowing those with consistently strong gradients to accumulate higher attention weights. Therefore, high attention scores emerge as learned signals of token importance through the optimization process.

To verify that attention-based selection identifies genuinely important tokens, we measured the MSE between pruned and dense model outputs across layers in Figure 8. The results confirm that tokens with higher attention weights have greater impact on model outputs.

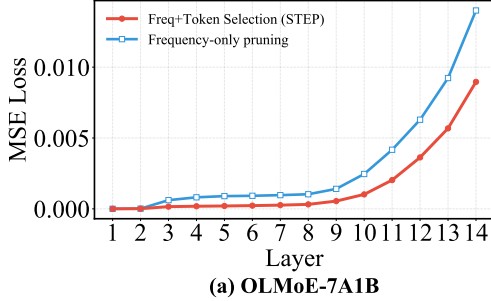

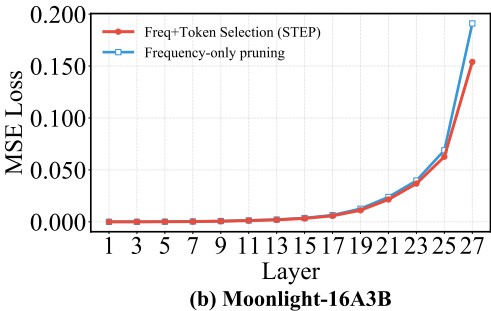

(a) OLMoE-7A1B

(b) Moonlight-16A3B

*Figure 8.* Per-layer MSE loss comparison. Frequency+Token Selection (STEP) vs. Frequency-only pruning. Lower is better.

Across all layers, Frequency + Token Selection achieves 7–75% lower MSE compared to frequency-only pruning. This indicates that our token importance selection correctly identifies critical experts whose removal has higher impact, whereas frequency-only pruning removes critical experts with larger immediate MSE and degrades downstream task performance. This validates that our selected tokens do matter for routing decisions and final predictions.

## D.2. Ablation on Alternative Indicators

We further compared attention-based selection against three alternative token importance metrics: gradient-based scoring ($||\nabla X_{in_i}||_2$), L2 norm-based scoring ($||X_{in_i}||_2$), and update magnitude scoring ($||X_{out_i} - X_{in_i}||_2$). Table 5 reports average performance across eight benchmarks under 50% and 20% token retention rates.

*Table 5.* Ablation on alternative token importance indicators.

| Method | No Selection | Attention | | Gradient | | L2 Norm | | Update Mag. | |
|---|---|---|---|---|---|---|---|---|---|
| | | 50% | 20% | 50% | 20% | 50% | 20% | 50% | 20% |
| Avg. Perf. | 56.16 | 58.42 | 58.81 | 56.58 | 56.40 | 53.10 | 51.62 | 53.44 | 53.07 |
| Gain | – | +2.26 | +2.65 | +0.42 | +0.24 | -3.06 | -4.54 | -2.72 | -3.09 |

*Table 6.* Performance comparison of different token selection methods integrated with our expert pruning framework on OLMoE-7A1B. where 50% and 20% denote the token retention rates

| Pruning Metric | No Selection | Selective Context | | LLMLingua | | Attention | |
|---|---|---|---|---|---|---|---|
| | | 50% | 20% | 50% | 20% | 50% | 20% |
| Frequency-based | 53.48 | 54.68 | 55.62 | 52.33 | 55.82 | 54.99 | 57.32 |
| Ours | 56.16 | 56.59 | 56.23 | 56.85 | 57.22 | 58.42 | 58.81 |

Comprehensive evaluation demonstrates that attention-based token selection is the most effective approach for expert pruning. This method achieves significant performance gains of 2.26 and 2.65 without additional computational overhead. In contrast, gradient-based scoring yields only marginal improvements of 0.42 and 0.24 while substantially increasing resource consumption. Methods relying on L2 norm or update magnitude notably degrade model performance. Further investigation confirms that token selection techniques from context compression literature integrate effectively into our pruning framework. We specifically evaluate Selective Context (Li et al., 2023) and LLMLingua (Jiang et al., 2023) under varying token retention rates. Both methods show consistent improvements as summarized in Table 6. Specifically, LLMLingua achieves competitive results with a score of 57.22 at 20% retention, whereas our attention-based method attains superior performance with a score of 58.81 under identical conditions. These findings validate our central thesis regarding the importance of token-aware expert evaluation. Among all evaluated metrics such as gradient-based, L2 norm-based, and update magnitude-based scoring, attention-guided selection provides the optimal balance between model effectiveness and computational efficiency.

# E. Additional Experiments on Model and Dataset Generalization

As shown in Table 7 and Table 8, we evaluate STEP on three large scale MoE architectures including GLM-4.5-Air (Team et al., 2025) and DeepSeek-MoE-16B-Base (Dai et al., 2024) under varying active expert counts. In addition to standard benchmarks, we include Chinese language and truthfulness datasets CMMLU, C-Eval, and TruthfulQA to assess cross lingual and safety aware generalization. Across all models and sparsity levels, STEP consistently outperforms MoNE, HC-SMoE, and GS, maintaining strong performance retention relative to the dense baselines. These results validate that STEP generalizes effectively to diverse large scale MoE architectures and task distributions.

# F. Additional Ablation Study on Key Hyperparameters

### F.1. Pruning Performance Analysis Across Expert Numbers

Figure 9 presents a comprehensive comparison of perplexity (PPL) results across different pruning methods with varying numbers of experts on both C4 and WikiText datasets. The expert count ranges from 64 to 20 (step size of 4), representing different levels of model sparsity. Our pruning method achieves superior performance preservation at high pruning ratios, significantly outperforming baseline approaches. At a 68.75% pruning ratio (reducing experts from 64 to 20), it reduces perplexity by approximately 98% compared to Gate Score, 72% compared to MONE, and 77% compared to HC-SMoE on WikiText, with similar improvements of 99% over Gate Score, 62% over MONE, and 83% over HC-SMoE on C4.

*Table 7.* Zero-Shot Task Accuracy(%) Comparison Across GLM-4.5-Air, DeepSeek-MoE-16B-Base, and Ling-mini-2.0 with Varying Active Expert Counts. Bold indicates best performance, underlined indicates second-best for each expert group.

| Model | Experts | Method | ARC-E | ARC-C | BoolQ | PIQA | Wino | Hella | MMLU | OBQA | Avg |
|-------|---------|--------|-------|-------|-------|------|------|-------|------|------|-----|
| GLM-4.5-Air | 128 | Dense | 85.82 | 60.32 | 88.10 | 82.70 | 77.90 | 67.56 | 78.90 | 48.40 | 73.71 |
| | 96 | GS | 82.41 | 54.44 | 86.54 | 81.88 | **77.43** | 65.78 | 62.25 | 47.00 | 69.72 |
| | | MoNE | 83.42 | 55.72 | 86.72 | 82.15 | 77.03 | **66.18** | 59.99 | **48.00** | 69.90 |
| | | **Ours** | **84.34** | **56.91** | **87.12** | **82.48** | 76.69 | 66.02 | **65.77** | 47.60 | **70.87** |
| | 64 | GS | 73.40 | 41.81 | 73.12 | 80.41 | 70.64 | 59.34 | 38.85 | 43.00 | 60.07 |
| | | MoNE | 75.76 | 45.39 | 79.72 | 80.41 | **75.30** | **61.37** | **41.95** | 43.40 | 62.91 |
| | | **Ours** | **76.32** | **46.43** | **80.14** | **80.79** | 75.08 | 60.55 | 48.03 | **44.00** | **63.92** |
| DeepSeek-MoE-16B-Base | 64 | Dense | 75.84 | 44.88 | 72.60 | 78.73 | 69.53 | 58.16 | 37.64 | 43.80 | 60.15 |
| | 48 | GS | 73.86 | 41.55 | 71.47 | **78.62** | 69.14 | **56.04** | 32.76 | 42.20 | 58.21 |
| | | MoNE | **75.59** | 43.16 | 73.06 | 78.45 | **69.46** | 55.95 | **33.14** | 44.00 | 59.10 |
| | | **Ours** | 75.53 | **43.42** | **74.25** | 78.40 | 69.06 | 56.12 | 32.63 | **44.60** | **59.25** |
| | 32 | GS | 65.68 | 33.19 | 65.61 | 74.86 | 62.35 | 48.37 | 22.99 | 37.80 | 51.35 |
| | | MoNE | 64.39 | 32.00 | 66.42 | **76.71** | **70.32** | **50.71** | **23.80** | **39.20** | 52.94 |
| | | **Ours** | **68.81** | **35.15** | **72.84** | 75.46 | 68.38 | 49.88 | 23.79 | 38.00 | **54.04** |

Performance degradation remains nearly linear and graceful as pruning intensity increases, while other methods exhibit exponential drops. This consistent superiority across datasets demonstrates robust generalization, with relative gains amplifying at higher pruning levels.

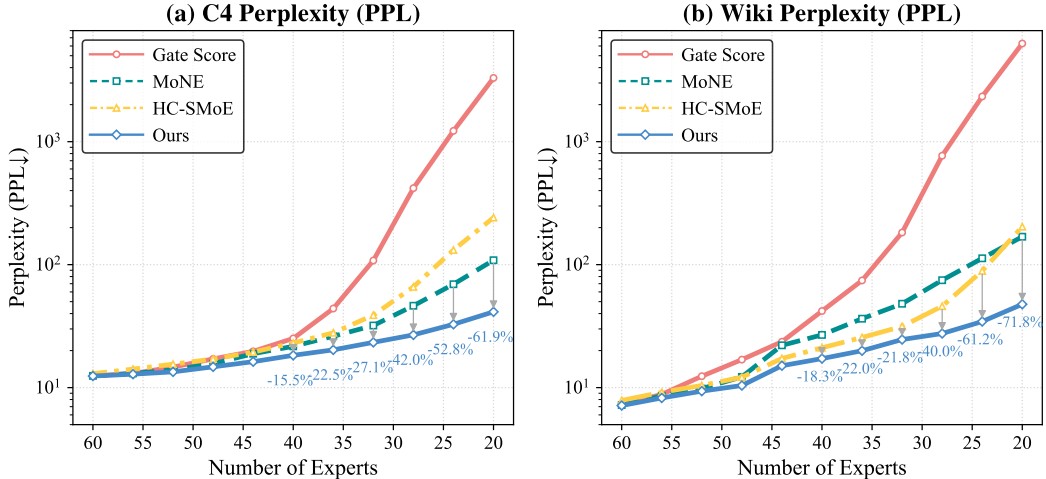

*Figure 9.* Perplexity of OLMoE-7A1B with varying expert numbers and pruning methods.

## F.2. Token Selection Ratio

Token selection ratio 0.5 is established as the default configuration, achieving optimal perplexity (12.58) while maintaining competitive zero-shot performance (58.42%), as detailed in Table 9. This ratio balances language modeling fundamentals with zero-shot task capabilities, yielding best-in-class results on ARC-C (44.03%) and OBQA (43.20%). Although ratio 0.2 yields a marginally higher zero-shot average (58.81%), it incurs a 0.8% relative perplexity degradation, favoring 0.5's superior stability-performance equilibrium.

## F.3. Experts-to-Bias Effectiveness.

To assess the universality of Stage 3, we compare performance with and without this component across multiple models and pruning ratios. Table 10 shows that Stage 3 consistently improves results under all conditions. Our full method achieves an

*Table 8.* Accuracy (%) on general tasks across different models and expert configurations. The reported average is computed over CMMLU, C-Eval, and TruthfulQA. Bold indicates the best performance within each expert group.

| Model | Experts | Method | CMMLU | C-Eval | TruthfulQA | Average |
|---|---|---|---|---|---|---|
| OLMoE-7A1B | 64 | Original | 34.67 | 33.89 | 37.38 | 35.31 |
| | 48 | Gate Score | 25.27 | 25.19 | 32.77 | 27.74 |
| | | MoNE | 26.67 | 25.84 | 34.46 | 28.99 |
| | | Ours | **27.99** | **29.05** | **35.15** | **30.73** |
| | 32 | Gate Score | 23.11 | 25.09 | 33.32 | 27.17 |
| | | MoNE | **26.89** | 25.09 | 35.13 | 29.04 |
| | | Ours | 26.20 | **26.30** | **35.69** | **29.40** |
| Moonlight-16A3B | 64 | Original | 78.05 | 76.52 | 45.88 | 66.82 |
| | 48 | Gate Score | 38.55 | 36.85 | 45.18 | 40.19 |
| | | MoNE | **45.60** | **47.03** | 45.53 | 46.05 |
| | | Ours | 45.32 | 46.15 | **47.90** | **46.46** |
| | 32 | Gate Score | 25.41 | 24.89 | 36.32 | 28.87 |
| | | MoNE | 25.44 | 25.11 | 37.12 | 29.22 |
| | | Ours | **25.52** | **25.56** | **44.92** | **32.00** |
| Qwen-30A3B | 128 | Original | 84.84 | 83.79 | 53.40 | 74.01 |
| | 96 | Gate Score | 52.39 | 55.27 | 47.93 | 51.86 |
| | | MoNE | 50.14 | 55.50 | **53.53** | 53.06 |
| | | Ours | **68.36** | **64.64** | 51.76 | **61.59** |
| | 64 | Gate Score | 35.92 | 38.26 | 46.13 | 40.10 |
| | | MoNE | 35.68 | 38.78 | 47.44 | 40.63 |
| | | Ours | **43.15** | **43.31** | **49.42** | **45.29** |

average score of 60.80, outperforming the variant without Stage 3 by 0.90 points at 59.90. This improvement holds across all tested models, including OLMoE, Moonlight, and Qwen3, at both sparsity levels of 25% and 50%. For fair comparison, all methods apply identical Stage 3 configurations with the same bias refinement procedure and calibration data. Even without Stage 3, our approach maintains strong performance at 59.90, surpassing both the baseline MoNE at 58.48 and MoNE with Stage 3 at 59.33, as well as all other methods in Table 1. This demonstrates that our core methodology delivers substantial gains independent of the calibration stage, with Stage 3 providing further refinement rather than compensating for inherent weaknesses.

### F.4. Bias Updating Hyperparameter Configuration

Table 11 details the hyperparameter specifications for bias updating. As shown in Table 12, perplexity decreases rapidly during the initial training phase, with the majority of gains achieved within the first epoch. Subsequent epochs yield diminishing returns, with identical results observed at epochs 3 (14.53/10.37) and 10 (14.43/10.98). This indicates model

*Table 9.* Ablation Study on Token Selection Ratio with 25% Expert Pruning on OLMoE-7A1B

| Ratio | Perplexity | | | Zero-shot Accuracy (%) | | | | | | | | |
|---|---|---|---|---|---|---|---|---|---|---|---|---|
| | Wiki | C4 | Avg | ARC-E | ARC-C | BoolQ | PIQA | Wino | Hella | MMLU | OBQA | Avg |
| 1.0 | 11.65 | 14.52 | 13.09 | 68.79 | 39.77 | 67.34 | 77.48 | 67.64 | 55.65 | 30.44 | 42.20 | 56.16 |
| 0.9 | 11.47 | 14.79 | 13.13 | 69.32 | 39.16 | 64.65 | 77.42 | 67.88 | 55.56 | 32.80 | 41.40 | 56.02 |
| 0.8 | 10.75 | 14.60 | 12.68 | 71.09 | 41.38 | 67.03 | 77.31 | **68.51** | 55.41 | 33.72 | 41.20 | 56.96 |
| 0.7 | 10.80 | 14.89 | 12.85 | 72.39 | 42.84 | 70.40 | 77.86 | 66.38 | 55.97 | 38.19 | 43.20 | 58.40 |
| 0.6 | 10.52 | 14.96 | 12.74 | 72.73 | 41.98 | 71.01 | 77.75 | 66.85 | 55.45 | 38.66 | 41.80 | 58.28 |
| 0.5 | 10.40 | **14.76** | **12.58** | 73.82 | **44.03** | 68.35 | 77.53 | 65.52 | 55.24 | 39.70 | **43.20** | 58.42 |
| 0.4 | 10.55 | 15.04 | 12.80 | 74.07 | 43.52 | 67.92 | 78.02 | 65.04 | 55.40 | 40.60 | 42.60 | 58.39 |
| 0.3 | 10.30 | 15.40 | 12.85 | 74.75 | 43.09 | 67.52 | **78.07** | 64.72 | 55.12 | 42.46 | 41.00 | 58.34 |
| 0.2 | **10.10** | 15.32 | 12.71 | **76.01** | 43.17 | 70.52 | 77.58 | 64.25 | 54.48 | **42.25** | 42.20 | **58.81** |
| 0.1 | 10.30 | 15.56 | 12.93 | 75.67 | 41.30 | 67.31 | 77.15 | 63.06 | 54.07 | **42.31** | 42.80 | 57.96 |

*Table 10.* Average performance on 8 zero-shot tasks with and without Stage 3 across different models and pruning ratios.

| Methods | OLMoE-7A1B | | Moonlight-16A3B | | Qwen3-30A3B | |
|---|---|---|---|---|---|---|
| | **25%** | **50%** | **25%** | **50%** | **25%** | **50%** |
| MoNE | 53.26 | 45.81 | 64.33 | 54.47 | 68.83 | 64.15 |
| Ours w/o Stage 3 | **57.51** | **48.31** | **64.97** | **54.82** | **68.84** | **65.05** |
| MoNE w/ Stage 3 | 54.08 | 47.55 | 64.51 | 55.78 | 69.16 | 64.92 |
| Ours w/ Stage 3 | **58.42** | **50.05** | **65.27** | **56.20** | **69.06** | **65.78** |

*Table 11.* Hyperparameter Specifications

| Parameter | Value |
|---|---|
| Optimizer | AdamW |
| Learning rate | $1 \times 10^{-4}$ |
| Weight decay | 0.1 |
| Epsilon ($\epsilon$) | $1 \times 10^{-6}$ |
| Precision | bfloat16 |
| Training steps | 128 |

*Table 12.* Perplexity by Training Epoch with 25% Expert Pruning on OLMoE-7A1B

| Epochs | C4 | Wiki |
|---|---|---|
| 0 | 15.52 | 12.87 |
| 1 | 14.76 | 10.40 |
| 3 | 14.53 | 10.37 |
| 10 | 14.43 | 10.98 |

convergence occurs early in training. We therefore adopt the 1-epoch configuration as the optimal balance between performance and computational efficiency for bias updating.

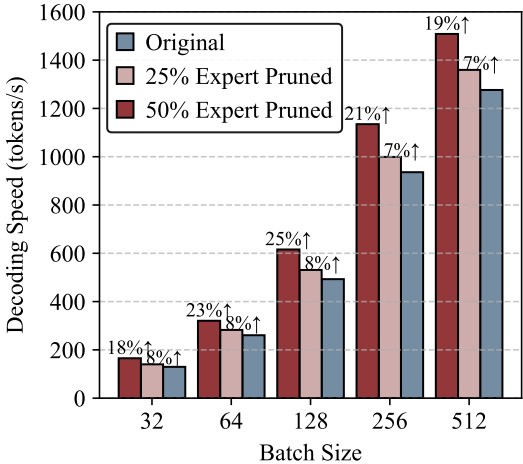

*Figure 10.* Throughput and Speedup of OLMoE-7A1B under different expert pruning ratios and different batch sizes.

## G. Model Inference Performance and Resource Consumption Analysis of OLMoE-7A1B

To further validate the efficiency of our pruning method across different architectures, we conduct experiments on the OLMoE-7A1B model. Figure 10 illustrates the throughput and speedup under varying expert pruning ratios and batch sizes, demonstrating consistent performance gains. Table 13 summarizes resource consumption and prefill performance, showing up to 46.6% memory reduction and 1.24× speedup with minimal pruning overhead.

## H. Attention Map across Different Layers

As demonstrated in Figure 11 and Figure 12, the token-to-token attention heatmap across layers in OLMoE-7A1B exhibits three key characteristics: (i) Sparse weight distribution with most attention weights below $10^{-3}$, (ii) Diagonal concentration of dominant weights, and (iii) Pronounced attention sink at initial token positions.

*Table 13.* Model Prefill Performance and Resource Consumption Analysis of OLMoE-7A1B.

| Model | Experts | Resource Consumption | | | | Prefill Performance | | Pruning Overhead | | |
| --- | --- | --- | --- | --- | --- | --- | --- | --- | --- | --- |
| | | Memory (GB) | | Parameters (B) | | Latency | Speedup | Time(mins) | | Memory |
| | | Value | ↓% | Value | ↓% | (ms) | | Prune | Bias | (GB) |
| **OLMoE-7A1B** | 64 | 12.91 | 0.0 | 6.92 | 0.0 | 43.07 | 1.00× | – | – | – |
| | 48 | 9.89 | 23.4 | 5.31 | 23.3 | 39.20 | 1.10× | 1 | 2 | 13.88 |
| | 32 | 6.90 | 46.6 | 3.70 | 46.5 | 34.69 | 1.24× | | | |

Quantitatively, for the last token's attention distribution, the top 20% tokens account for $\sim 80\%$ of cumulative attention weight, with layer-wise variations:

- **Layer 16**: Top 20% tokens account for 92.7% attention weight

- **Layer 12**: Top 20% tokens account for 93.8% attention weight

- **Layer 8**: Top 20% tokens account for 93.4% attention weight

- **Layer 4**: 75.5% concentration, the minimum observed value

These findings underscore that optimal expert pruning strategies must account for layer-wise variations in token selection dynamics.

## I. Visual Analysis of Routing Distributions Before and After Expert Pruning

We compare the expert load across C4, Wikitext2, and Arc-challenge tasks. As clearly shown in the Figure 13 and Figure 14, after expert pruning, the routing patterns consistently exhibit minimal divergence, demonstrating that our bias updating strategy effectively compensates for expert removal without destabilizing the routing mechanism.

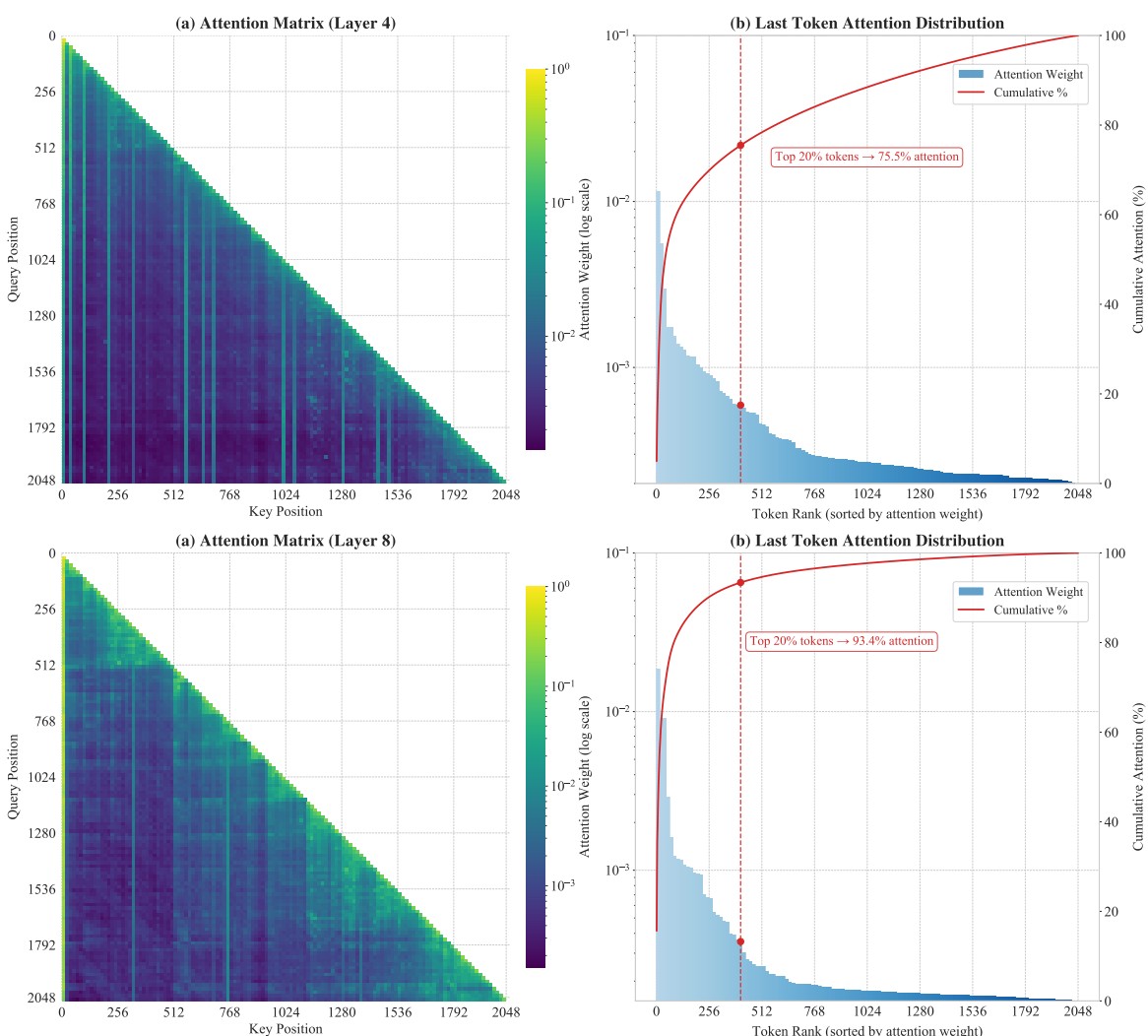

*Figure 11.* Attention map and last-token attention distribution visualization for Layers 4/8 (of 16) in OLMoE-7A1B.

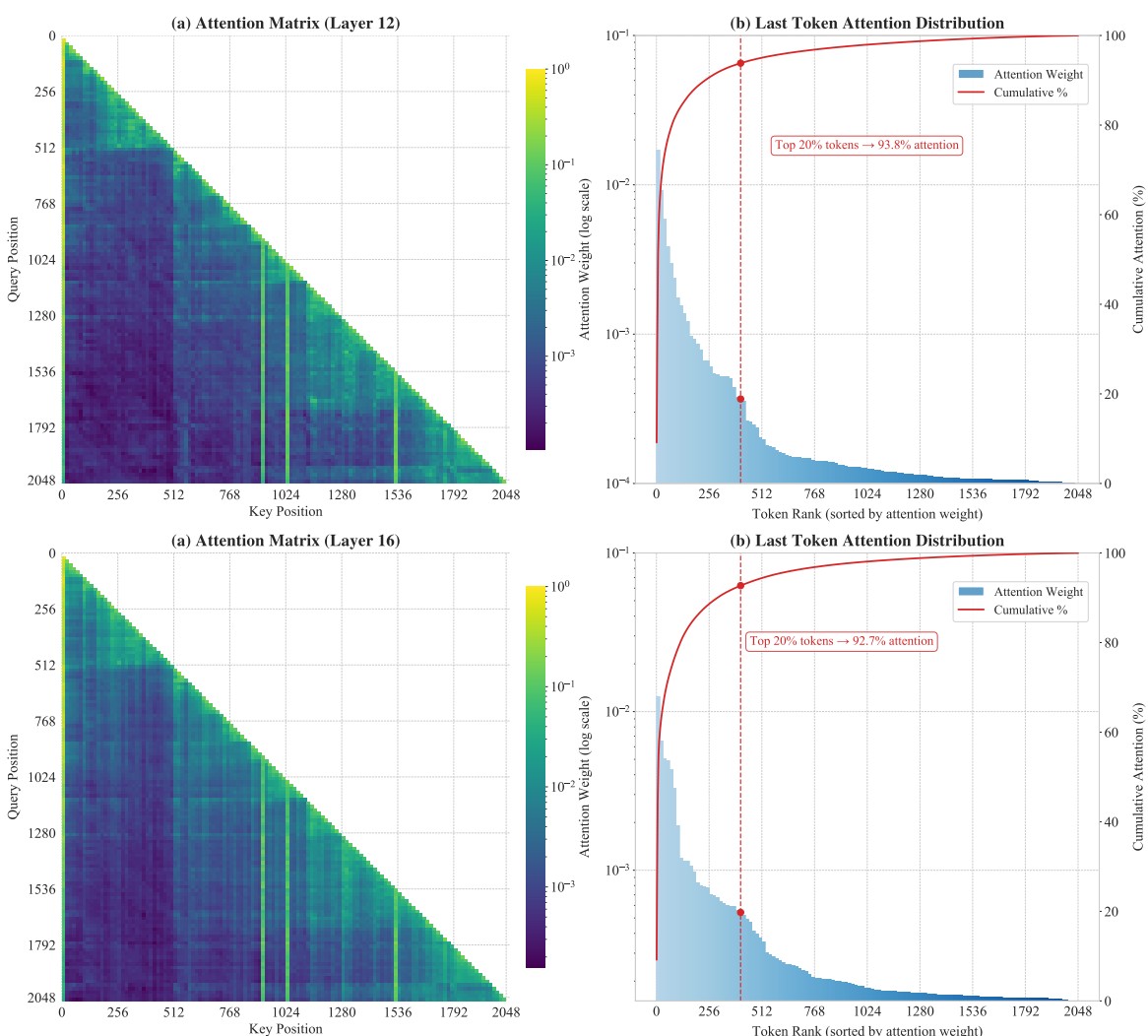

*Figure 12.* Attention map and last-token attention distribution visualization for Layers 12/16 (of 16) in OLMoE-7A1B.

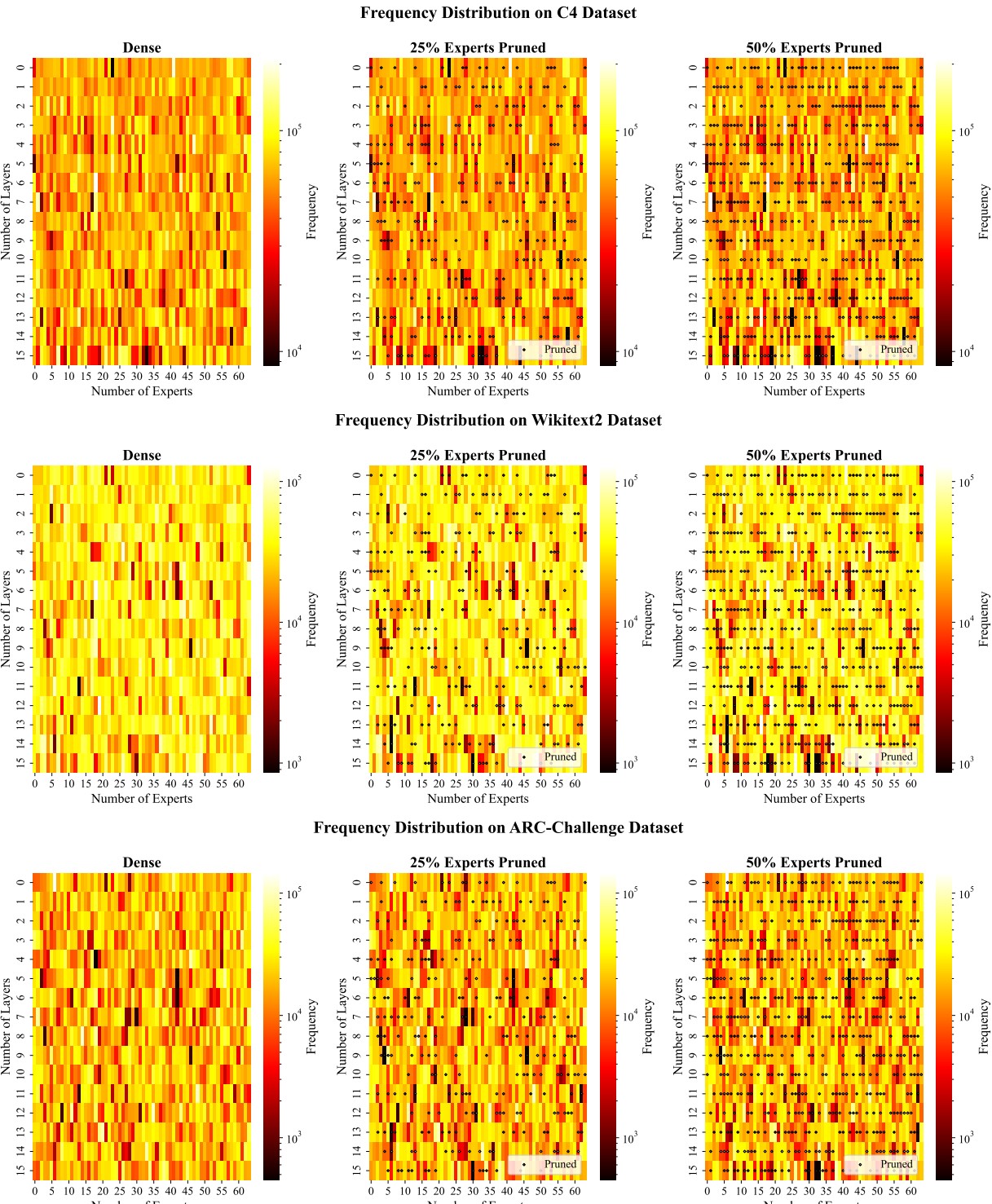

*Figure 13.* Expert routing frequency distribution of OLMoE-7A1B across C4, Wikitext2 and ARC-Challenge dataset.

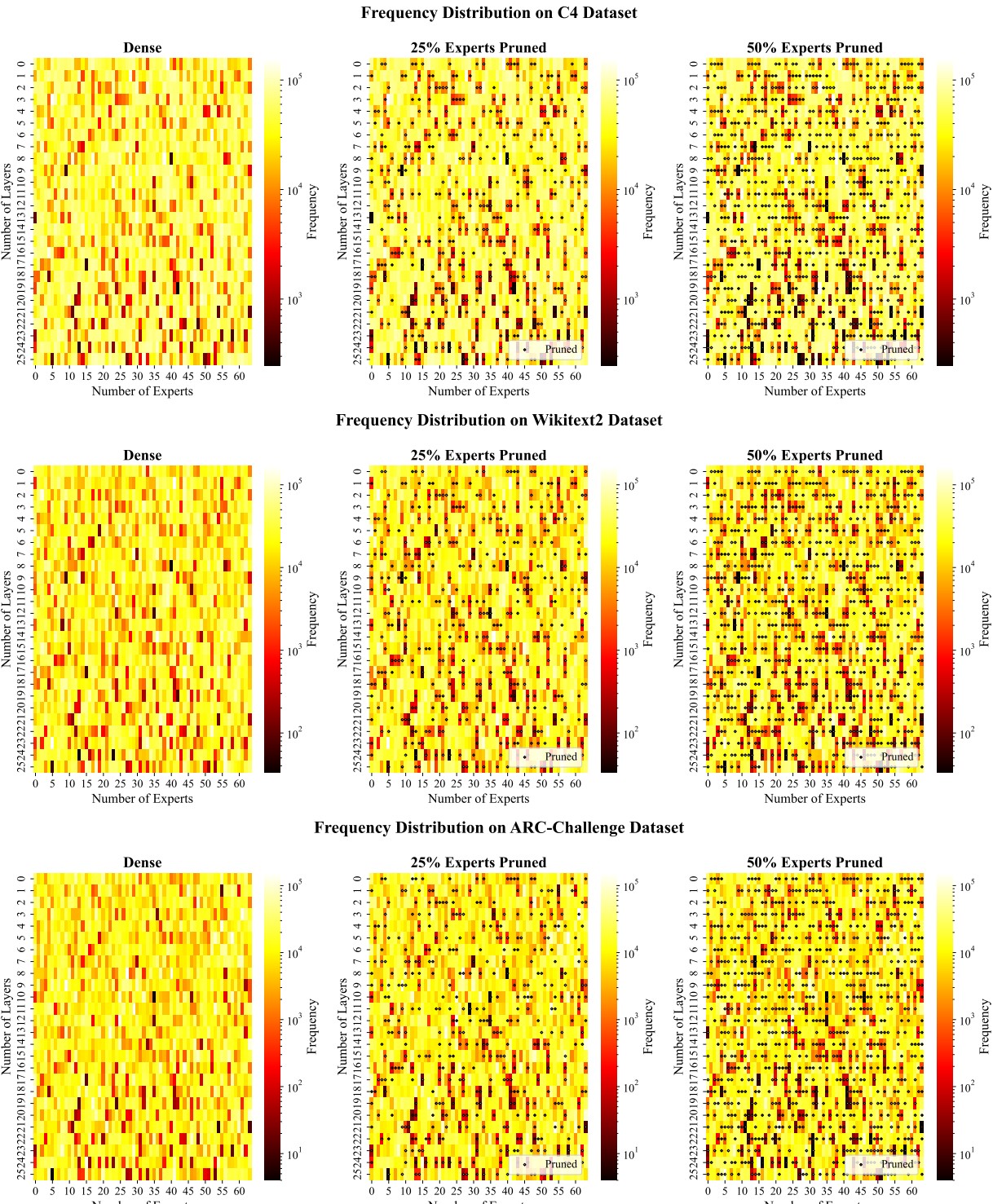

*Figure 14.* Expert routing frequency distribution of Moonlight-16A3B across C4, Wikitext2 and ARC-Challenge dataset.

