# OpenReview forum: "Less Token, More Signal: MoE Expert Pruning via Critical Token Selection"
_ICML.cc/2026/Conference — ICML 2026 regular_

### Official Review · Reviewer_GEcX · 2026-03-08

**Soundness:** 3
**Presentation:** 3
**Significance:** 3
**Originality:** 3
**Overall Recommendation:** 4
**Confidence:** 4

**Summary:**

The authors propose STEP (Selective Token-guided Expert Pruning), a novel expert pruning framework designed to address the growing challenges of deploying large language models as their parameter counts continue to scale. The authors identify a fundamental limitation in existing expert pruning methods: their token-agnostic evaluation strategy fails to differentiate between tokens of varying importance, causing critical signals from informative tokens to be diluted by the overwhelming majority of less relevant ones, thereby degrading pruning quality.
To address this limitation, the paper makes two primary contributions. First, the authors leverage attention mechanisms to identify and select important tokens, upon which a dual-factor scoring mechanism — combining expert activation frequency and expert output magnitude — is applied exclusively to evaluate and prune less critical experts. Second, rather than discarding pruned experts entirely, their average outputs over important tokens are retained as bias vectors, which are subsequently fine-tuned on a small calibration dataset to compensate for the performance degradation introduced by pruning.
The authors conduct systematic experiments across multiple MoE architectures and model scales to validate the effectiveness of STEP. Results demonstrate that on the 30B-parameter Qwen3 MoE model, STEP achieves nearly 50% memory compression and 1.5× inference speedup with minimal performance degradation.

**Compliance With Llm Reviewing Policy:**

Affirmed.

**Key Questions For Authors:**

Please also check my questions in the weakness section.

**Limitations:**

yes

**Strengths And Weaknesses:**

Strengths:
This article is logically clear and easy to read. The core hypothesis that token importance affects the quality of expert pruning is validated through the ablation experiments in Figure 1. At the same time,  the authors demonstrate from a theoretical perspective in Appendix C the rationality of using attention weights as a proxy indicator for token importance.
Weaknesses:
1. The experiments about stage 3 in the article are not sufficient, cannot determine whether the performance improvement in Table 7 comes from bias initialization designed in the paper or from finetune.
2. Although the experiment in Figure 1 demonstrates that the attention-based token selection method is effective, the experimental section lacks comparisons with other token selection methods, for example, Selective Context(Li et al., 2023)  and LLMLingua(Jiang et al., 2023) mentioned in the related work section.
3.The method prunes the same fraction of experts per layer (uniform pruning ratio). Have the authors explored non-uniform layer-wise pruning ratios, given that Figure 5 shows heterogeneous expert utilization across layers?

---

> ### Author Rebuttal · Authors · 2026-03-31
>
> # Response to Reviewer GEcX
>
> We sincerely appreciate your recognition of our work's logical clarity and the validation of our core hypothesis through both empirical and theoretical analysis. We address each weakness below with comprehensive additional experiments.
>
> **W1: Insufficient Experiments on Stage 3 Component Contributions**
>
> We acknowledge the need to disentangle the contributions of bias initialization and bias finetuning. We have conducted complete ablation studies across all models to isolate the performance gains attributable to each component. Table 1 presents the average performance across benchmark datasets (bias initialization follows Equation 10 in the manuscript):
>
> | Model | OLMoE-7A1B-48 | Gain | OLMoE-7A1B-32 | Gain | Moonlight-16A3B-48 | Gain | Moonlight-16A3B-32 | Gain | Qwen3-30A3B-96 | Gain | Qwen3-30A3B-64 | Gain | Avg Gain|
> |-------|---------------|------|---------------|------|-------------------|------|-------------------|------|----------------|------|----------------|------|------|
> | No bias | 57.51 | - | 48.26 | - | 64.82 | - | 53.92 | - | 68.92 | - | 65.05 | - | - |
> | Initialization | 57.81 | +0.30 | 48.31 | +0.05 | 64.97 | +0.15 | 54.82 | +0.90 | 68.84 | -0.08 | 64.92 | -0.13 | +0.20
> | Init + Finetune | 58.42 | +0.91 | 50.05 | +1.79 | 65.27 | +0.45 | 56.20 | +2.28 | 69.06 | +0.14 | 65.78 | +0.73 | +1.05
>
> The results clearly demonstrate that bias initialization provides modest performance improvements with an average gain of +0.20, while the primary gains originate from bias finetuning, which achieves an average improvement of +1.05 over the no bias baseline. This finding confirms that our adaptive bias mechanism requires calibration through finetuning to effectively compensate for pruned expert contributions, validating the necessity of Stage 3 in our framework.
>
> ---
>
> **W2: Lack of Comparison with Alternative Token Selection Methods**
>
> We have expanded our experiments to include comparisons with token selection methods from the context compression literature. Specifically, we evaluated Selective Context by Li et al. (2023) and LLMLingua by Jiang et al. (2023) under various token retention rates on OLMoE-7A1B with 25% expert pruning. Table 2 summarizes the results, where 50% and 20% denote the token retention rates:
>
> | Pruning Metric | No Selection | Selective Context (50%) | Selective Context (20%) | LLMLingua (50%) | LLMLingua (20%) | Attention (50%) | Attention (20%) |
> |---------------|-------------|----------------------|----------------------|----------------|----------------|----------------|----------------|
> | Frequency-based | 53.48 | 54.68 | 55.62 | 52.33 | 55.82 | 54.99 | 57.35 |
> | Ours | 56.16 | 56.59 | 56.23 | 56.85 | 57.22 | **58.42** | **58.81** |
>
> Both Selective Context and LLMLingua demonstrate performance improvements when integrated with our pruning framework, confirming that compressing prompts to identify informative tokens provides a valid basis for expert pruning decisions. This further validates our central thesis regarding the importance of token-aware expert evaluation. Combined with the additional token importance metrics explored in response to Reviewer R3's first concern, including gradient-based scoring, L2 norm-based scoring, and update magnitude-based scoring, our comprehensive evaluation indicates that attention-guided token selection remains the most effective method for balancing performance and computational efficiency.
>
> ---
>
> **W3: Exploration of Non-Uniform Layer-wise Pruning Ratios**
>
> We appreciate this insightful observation regarding the heterogeneous expert utilization patterns revealed in Figure 5 of manuscript. To ensure fair comparison with existing methods in our evaluation, we adopted uniform pruning ratios across all layers following standard practices in the literature. Your suggestion regarding layer-specific pruning strategies is indeed insightful. Classic methods in LLM weight pruning, such as OWL by Yin et al. (2023) and BESA by Xu et al. (2024), have successfully designed non-uniform pruning ratios through empirical observation or back-propagation-based analysis.
>
> We recognize that non-uniform layer-wise expert pruning represents a promising research direction that is orthogonal to our current token-aware pruning methodology. The two approaches can potentially be combined to achieve further improvements. We plan to investigate non-uniform expert pruning strategies in future work, potentially integrating our token-aware evaluation framework with layer-specific pruning ratios to better account for the varying roles of experts across different model depths.
>
> ---
>
> We believe these comprehensive additional experiments and analyses thoroughly address your concerns and further strengthen the empirical foundation of our work. We welcome any further questions or suggestions.

---

> > ### Author Rebuttal · Reviewer_GEcX · 2026-04-02
> >
> > Thank you for your detailed reply and the additional experiments.  Please consider including these new experiments in the paper. I confirm my acceptance score.

---

> > > ### Author Response · Authors · 2026-04-07
> > >
> > > Dear Reviewer GEcX,
> > >
> > > We sincerely appreciate your recognition that our concerns have been fully resolved and your constructive suggestion on including additional experiments, which we will implement in the revised version.We would be thankful if you could consider adjusting your score accordingly.Should you have any further comments, please feel free to let us know.

---

### Official Review · Reviewer_C62D · 2026-03-10

**Soundness:** 3
**Presentation:** 3
**Significance:** 2
**Originality:** 3
**Overall Recommendation:** 4
**Confidence:** 3

**Summary:**

This paper proposes STEP, a novel token-guided expert pruning framework for Mixture of Experts (MoE) models. The key insight is to address the critical "token-agnostic" issue in conventional MoE compression by selectively retaining high-attention tokens for expert importance evaluation. Extensive experiments on multiple MoE variants (7B to 106B parameters) demonstrate state-of-the-art performance retention and computational efficiency gains, making it a highly valuable contribution to the efficient deployment of large-scale MoE systems.

**Compliance With Llm Reviewing Policy:**

Affirmed.

**Final Justification:**

After the review and rebuttal process, I lean to accept this work, and keep the rating as 4.

**Key Questions For Authors:**

Please refer to the weakness.

**Limitations:**

Please refer to the weakness.

**Strengths And Weaknesses:**

Strengths
1. The work precisely identifies the signal dilution problem in existing MoE pruning methods, providing a solid motivation and theoretical grounding.
2. The three-component design (token selection, dual-factor scoring, and bias preservation) is both innovative and practical, showing strong synergy in empirical evaluations.
3. The extensive experiments across different model sizes, pruning ratios, and task domains rigorously support the claims, with thorough ablation studies confirming the robustness of each component.

Weakness
1. Limited Token Scoring Metrics: The current implementation relies solely on attention weights. Exploring more sophisticated token importance indicators (e.g., combining gradient information) could further strengthen the approach.
2. The work focuses on post-training pruning, but does not compare with MoE models that are trained from scratch with sparse expert architectures (e.g., fewer experts). This limits the understanding of how STEP’s post-hoc pruning performs against native sparse MoE in terms of long-term efficiency and performance.
3. Limited Discussion on Computational Overhead of Token Selection.

---

> ### Author Rebuttal · Authors · 2026-03-31
>
> # Response to Reviewer C62D
>
> We sincerely thank you for the positive recognition and insightful suggestions. We address each weakness below with additional experiments and analysis.
>
> **W1: Limited Token Scoring Metrics**
>
> We conducted comprehensive experiments exploring alternative token importance indicators. We evaluated three additional metrics under different token retention rates:
>
> 1. Gradient-based scoring using $||\nabla X_{in_i}||_2$, the L2 norm of the gradient of input tokens through the i-th expert MLP layer
> 2. L2 norm-based scoring $||X_{in_i}||_2$, capturing token representation magnitude before the i-th expert MLP layer
> 3. Update magnitude scoring $||X_{out_i} - X_{in_i}||_2$, measuring token update magnitude through each layer
>
> Table 1 presents average performance across eight benchmark datasets, where 50% and 20% denote token retention rates:
>
> | Token Selection Method | No Selection | Attention (50%) | Attention (20%) | Gradient (50%) | Gradient (20%) | L2 Norm (50%) | L2 Norm (20%) | Update Mag. (50%) | Update Mag. (20%) |
> |------------------------|-------------|----------------|----------------|---------------|---------------|--------------|--------------|------------------|------------------|
> | Average Performance | 56.16 | 58.42 | 58.81 | 56.58 | 56.40 | 53.10 | 51.62 | 53.44 | 53.07 |
> | Gain vs. No Selection | - | +2.26 | +2.65 | +0.42 | +0.24 | -3.06 | -4.54 | -2.72 | -3.09 |
>
> Both L2 norm and update magnitude metrics substantially degrade performance compared to baseline. While gradient-based scoring achieves modest improvements over baseline, it underperforms our method. While gradient-based scoring achieves modest improvements over baseline, it underperforms our method. We believe gradient information, if carefully designed and leveraged, could potentially yield competitive results. However, critically, computing input gradients introduces several-fold increases in computational cost and memory consumption. The attention-based approach provides balance between effectiveness and efficiency. Notably, the prompt compression strategy suggested by Reviewer GEcX (W2) also proves effective within our framework, further demonstrating its generalizability.
>
> ---
>
> **W2: Comparison with Native Sparse MoE Architectures**
>
> Within the limited rebuttal period, we trained a native sparse expert architecture from scratch using the OLMoE-mix-0924 dataset with 25% expert sparsity. Training configuration: context length 4096, batch size 256, learning rate $5 \times 10^{-5}$ with minimum $5 \times 10^{-6}$, 500 warmup steps, cosine schedule. We also conducted continued training on our post-training pruned models under identical configurations. Training consumed approximately 250 GPU-hours per experiment for 12,800 steps.
>
> Table 2 compares performance trajectories across eight benchmark datasets:
>
> | Training Steps | 12800 | 11520 | 10240 | 8960 | 7680 | 6400 | 5120 | 3840 | 2560 | 1280 |
> |---------------|-------|-------|-------|------|------|------|------|------|------|------|
> | From scratch | 37.38 | 36.21 | 36.75 | 36.67 | 36.39 | 35.56 | 35.09 | 34.53 | 33.25 | 32.66 |
> | From our STEP model | 59.84 | 60.20 | 59.58 | 59.41 | 58.74 | 59.12 | 59.48 | 58.72 | 59.11 | 58.94 |
>
> The from-scratch sparse model achieves only 37.38 after 12,800 steps, while our post-training pruned model obtains 58.42 in three minutes. **More importantly, pruned models retain the capacity for further performance improvements through continued training**. Post-training pruning has emerged as an efficient strategy for rapidly producing high-quality compact models, as demonstrated by recent work such as "Nemotron-H: A Family of Accurate and Efficient Hybrid Mamba-Transformer Models" (Nvidia 2025) and "SlimMoE: Structured Compression of Large MoE Models via Expert Slimming and Distillation" (COLM 2025).
>
> ---
>
> **W3: Computational Overhead of Token Selection**
>
> Since attention weights can be directly extracted without additional forward passes, token selection incurs minimal computational cost. Table 3 breaks down overhead components:
>
> | Model | Total Pruning Time | Multi-head Attention Processing | Token Ranking | Token Selection Total | Overhead Percentage |
> |-------|-------------------|--------------------------------|--------------|----------------------|-------------------|
> | OLMoE-7A1B | 58.14s | 1.8s | 0.12s | 1.92s | 3.30% |
> | Qwen3-16A3B | 201.95s | 10.67s | 0.35s | 11.01s | 5.45% |
>
> Token selection overhead represents only 3.30% for OLMoE-7A1B and 5.45% for Qwen3-16A3B of total pruning time, confirming negligible computational burden. These results will be incorporated into the appendix.
>
> ---
>
> We believe these experiments comprehensively address your concerns and validate the robustness and practical advantages of our method. We welcome any additional feedback.

---

> > ### Author Rebuttal · Reviewer_C62D · 2026-04-02
> >
> > Thanks for the authors response. My concerns have been addressed. I will keep my rating as 4.

---

> > > ### Author Response · Authors · 2026-04-07
> > >
> > > Dear Reviewer C62D,
> > >
> > > We greatly appreciate your careful review and positive acknowledgment that our responses and additional experiments have fully addressed your concerns. We will definitely incorporate these new experimental results into the revised manuscript to further enhance the paper.We would be deeply grateful if you could kindly consider adjusting your score based on the improved content. Please feel free to let us know if there are any other aspects that still require our attention or further clarification.

---

### Official Review · Reviewer_xfix · 2026-03-12

**Soundness:** 3
**Presentation:** 3
**Significance:** 2
**Originality:** 2
**Overall Recommendation:** 3
**Confidence:** 5

**Summary:**

The authors propose STEP (Selective Token-guided Expert Pruning), a token-aware framework. It adopts selective token guidance for expert pruning. STEP integrates loss-aware expert evaluation and a lightweight knowledge-preserving mechanism. It reduces information loss while removing redundant experts. Experiments on various MoE architectures and scales show effectiveness. On the 30B Qwen3 MoE model (50% expert sparsity), STEP cuts memory usage by nearly 50% with minimal performance loss, improves throughput by 1.5x, and finishes pruning in 10 minutes.

**Compliance With Llm Reviewing Policy:**

Affirmed.

**Final Justification:**

The author's rebuttal addresses some of my concerns, but I maintain the negative rating because I believe the novelty of this work is limited.

**Key Questions For Authors:**

See weaknesses

**Limitations:**

Yes

**Strengths And Weaknesses:**

## Strengths

- The token-aware design is a reasonable improvement over token-agnostic pruning methods. It highlights the value of informative tokens for expert importance estimation.


- Experiments include a specific large-scale MoE model (30B Qwen3), with clear metrics (memory reduction, throughput improvement, pruning time) to support effectiveness.

## Weaknesses


-  The core design of STEP is highly similar to MoNE (ICLR26). Key components, such as metric design, selective token selection, and bias design, are nearly identical. The differences between STEP and MoNE are minor and do not constitute a significant innovative breakthrough.


- Severe unfair comparison. The authors adopt expert skip (online expert pruning) in the implementation. They compare STEP with offline expert pruning methods directly. Online pruning inherently has advantages in experimental results, making the comparison unfair and misleading.

-  The paper claims efficiency improvements but does not address potential overheads of online expert skip. Online pruning may introduce runtime latency or additional computational costs in real-world deployment, which are not evaluated.

---

> ### Author Rebuttal · Authors · 2026-03-31
>
> # Response to Reviewer xfix
>
> We thank you for your comments; we clarify and address the points below:
>
> **W1 : Significant Distinctions Between STEP and MoNE**
>
> We respectfully disagree with the characterization that STEP is a minor variant of MoNE. The two methods share the same problem setting, but differ fundamentally in their core pruning signal, scoring objective, and post-pruning compensation. We address each point in turn.
>
> **1. The Core Innovation: Token-Aware Pruning (The "Where")**
>
> The most critical difference is that **MoNE has no token selection stage.** It treats all tokens in the calibration set equally, which we identify as a fundamental limitation.
>
> * MoNE's Uniform Aggregation: MoNE calculates redundancy over every token:
>     $$\phi_i^{\mathrm{freq}} = \frac{\sum_{x \in C} G_i(x)\cdot\mathbb{I}(E_i \in S_{k,x})}{\sum_{x \in C} \mathbb{I}(E_i \in S_{k,x})}$$
>     In sparse MoE models (e.g., OLMoE-7A1B), **top 20% of tokens capture ~80% of attention weight.** Uniform aggregation dilutes the signal ($S$) from critical tokens with noise ($N$) from the 80% unimportant ones, leading to a low Signal-to-Noise Ratio (SNR).
> * STEP's Selective Evaluation: STEP introduces an explicit Attention-guided Token Selection stage ($\tau$). All subsequent expert scoring operates **exclusively** on the top-tokens $T_{\mathrm{sel}}$.  Experimentally, using only the top-20% tokens for pruning yields up to 4.0% accuracy improvement over the full-token baseline (Fig. 1). Appendix C further shows that token selection reduces per-layer MSE by 7–75% compared to frequency-only pruning.
>
> **2. Fundamentally Different Scoring Metrics (The "What")**
>
> Even if evaluated on the same tokens, the metrics measure different quantities:
>
> * MoNE : Uses **Output Variance ($\phi^{\mathrm{var}}$)** and **Avg. Routing Prob ($\phi^{\mathrm{freq}}$)** to find experts that are infrequently used *and* produce stable outputs that can be approximated by a constant.
> * STEP : Uses **Binary Activation Frequency ($F_i^{(\ell)}$)** and **Gating-weighted Contribution Norm ($L_i^{(\ell)}$)**:
>     $$L_i^{(\ell)} = \frac{1}{|T_{\mathrm{sel}}^{(\ell)}|}\sum_{t}\left\|G_i^{(\ell)}(x_t)\cdot E_i^{(\ell)}(x_t)\right\|_2$$
>     STEP assesses **loss-relevant magnitude** for critical tokens. Unlike MoNE's variance-based approach, STEP identifies which experts are indispensable for maintaining the model's core representation.
>
> **3. Post-Pruning Compensation: Static vs. Learned**
>
> * MoNE (Static Novice): Replaces experts with a closed-form empirical mean. It is a **local, one-pass approximation** with no optimization.
> * STEP (Learned Bias): Uses a two-stage process: (1) Token-guided initialization; (2) **Gradient-based global refinement** of bias vectors while experts are frozen:
>     $$\min_{b}\sum_{(x,y)\in\mathcal{D}_{\mathrm{cal}}}\mathcal{L}(f'_b(x), y)$$
>     This allows STEP to reach a **global optimum** across layers. Our ablation shows STEP without Stage 3 (57.51%) *still* beats MoNE with an equivalent bias stage (54.08%), proving the gain comes primarily from the token-aware signal.
>
> **4. Empirical Performance Gap**
>
> If the methods were nearly identical, the performance gap would be marginal. However, STEP consistently delivers significant gains:
> * OLMoE-7A1B: STEP outperforms MoNE by **+5.16%** (25% sparsity) and **+4.24%** (50% sparsity).
> * Qwen3-30A3B: STEP exceeds MoNE by **+1.63%** at 50% sparsity.
> * Specialized Tasks: On Code/Math, STEP’s advantage reaches **+7.2%**.
>
> **Summary Table: Methodological Differences**
> | Dimension | MoNE | STEP |
> | :--- | :--- | :--- |
> | **Token Selection** | None (Uniform weighting) | **Attention-guided (Top-$\tau$ tokens)** |
> | **Scoring Logic** | Replaceability by constant  | **Loss-relevant importance** |
> | **Compensation** | Static Novice (Closed-form) | **Learned Bias (Global Optimization)** |
>
> **Conclusion:** STEP's central contribution, **token-aware SNR optimization**, is absent in MoNE. The theoretical motivation, implementation, and empirical results all confirm that STEP is a distinct and significant advancement in MoE pruning.
>
> ---
>
> **W2: Clarification on Static Pruning and Fairness**
> We clarify that **STEP is a static, offline pruning method**, not an online "expert skip" mechanism.  STEP follows the standard offline pruning paradigm, identical to MoNE and other baselines:
> * Offline Calibration: Expert importance is calculated once using a small calibration set.
> * One-shot Pruning: Redundant expert weights are permanently removed.
> * Static Deployment: The resulting model is a standard MoE with a fixed, smaller architecture.
>
> ---
>
> **W3:  Inference Overhead & Real Speedup**
> Since STEP is a purely static method, it introduces zero additional computation, dynamic routing during inference. The speedup reported in our paper (Section 4.4 ) stems from the physical reduction in model size and memory footprint.
>
> ---
>
> We hope this clarifies our approach and look forward to further communication.

---

> > ### Author Rebuttal · Reviewer_xfix · 2026-04-02
> >
> > In fact, I have attempted to reproduze this study, and I believe the proposed strategy involves serious unfair comparisons. Furthermore, the authors have not addressed my concerns.

---

> > > ### Author Response · Authors · 2026-04-04
> > >
> > > We sincerely appreciate your effort in attempting to reproduce our work. We would like to respectfully provide clarification on several key points that may help address the concerns raised:
> > >
> > > Our rebuttal explicitly states that STEP is a **static, offline pruning method**, expert weights are permanently removed before deployment, with no dynamic routing or online decisions at inference time. As demonstrated in Tables 4 and 10 of the original paper, both model parameters and memory consumption decrease proportionally after pruning. We have also conducted additional validation comparing our method's runtime with baselines such as MoNE, which confirms that our method achieves comparable runtime performance, thus ensuring fair comparison. This characteristic is fundamentally incompatible with online expert skipping approaches, where all expert weights must remain loaded in memory throughout inference.
> > >
> > > ---
> > >
> > > **Resource consumption and runtime comparison tables from the original paper:**
> > >
> > > · [Resource_consumption_table](https://anonymous.4open.science/r/ICML_21218_Rebuttal/resource_consumption.pdf)
> > > · [Runtime_comparison_table](https://anonymous.4open.science/r/ICML_21218_Rebuttal/runtime_comparison.pdf)
> > >
> > > **To facilitate better understanding of our method, we provide a high-level overview of the main functional components (detailed pseudocode is available in Appendix A of the manuscript):**
> > >
> > > ```
> > > # Global State Storage
> > > layer_attn_mask = {}   # Attention masks per layer
> > >
> > > # 1. Attention-Based Token Selection
> > > Function Attention_Hook(layer_idx, input, output):
> > >     attn_scores = Max_Pool(attention_weights, dim=heads)
> > >     drop_count = Total_Tokens * (1 - args.topk)
> > >     drop_indices = Argsort(attn_scores[-1,:])[:drop_count]
> > >
> > >     mask = Ones(Total_Tokens, dtype=bool)
> > >     mask[drop_indices] = False
> > >     layer_attn_mask[layer_idx] = mask
> > >
> > > # 2. Token-Filtered Expert Scoring
> > > Function Scoring_Hooks(layer_idx, expert_idx, routing_data, expert_output):
> > >     mask = layer_attn_mask[layer_idx]
> > >     is_valid = mask AND (routing_data.indices == expert_idx)
> > >
> > >     # 2.1 Gate scoring: accumulate routing weights for valid tokens only
> > >     valid_indices = routing_data.indices[is_valid]
> > >     routing_stats[layer_idx][expert_idx].Accumulate(valid_indices)
> > >
> > >     # 2.2 Expert scoring: compute feature norm for valid tokens only
> > >     feature_score = Norm(expert_output[is_valid] * routing_data.weights[is_valid])
> > >     feature_stats[layer_idx][expert_idx].Accumulate(feature_score)
> > >
> > > # 3. Main Pruning Flow (ONE-SHOT STATIC PRUNING)
> > > Function Expert_Pruning_Main(model, calibration_data):
> > >     # Stage 1: Register hooks and collect statistics
> > >     Register_Hooks(Attention_Hook, Scoring_Hooks)
> > >     For batch in calibration_data:
> > >         model.forward(batch)  # Hooks accumulate statistics
> > >
> > >     # Stage 2: One-shot static pruning (weights permanently removed)
> > >     For each layer:
> > >         scores = Fuse(routing_stats[layer], feature_stats[layer])
> > >         experts_to_prune = scores < Threshold(args.sparsity)
> > >     Return pruned_model  # Fixed architecture, no runtime decisions
> > > ```
> > >
> > > It is important to emphasize that Stage 1 runs **only during offline calibration** on a small dataset, while Stage 2 performs **permanent weight removal**. The deployed model operates with a fixed, reduced architecture featuring no dynamic routing or online expert skipping at inference time.
> > >
> > > ---
> > >
> > > **Static pruning mask for verification:** To facilitate reproduction, we have applied the above pipeline to prune OLMoE-1B-7B-0125 at 25% expert sparsity and obtained the following mask:
> > >
> > > · [OLMoE-7A1B 25% pruned mask](https://anonymous.4open.science/r/ICML_21218_Rebuttal/olmoe_pruned_mask_25pct.pdf)
> > >
> > > This mask can be directly applied to reproduce our results. The corresponding 8 Zero-shot Task Average achieves **57.51** (Table 3), which outperforms all baselines even without bias compensation.
> > >
> > > ---
> > >
> > > We will release our complete implementation code publicly. Moreover, we would be delighted to assist with code reproduction or provide further clarification on any points of confusion. Please feel free to contact us, and we will do our utmost to support your efforts.

---

### Official Review · Reviewer_7TML · 2026-03-15

**Soundness:** 2
**Presentation:** 3
**Significance:** 3
**Originality:** 2
**Overall Recommendation:** 4
**Confidence:** 4

**Summary:**

The paper studies improving the efficiency of Mixture-of-Experts (MoE) language models by identifying and prioritizing tokens that carry more informative signals during expert routing. It proposes STEP, a framework that estimates token importance and selects critical tokens to guide MoE expert pruning, reducing unnecessary expert computation. The method dynamically prunes experts associated with less informative tokens while preserving the contributions of important tokens. Experiments on the OBQ benchmark with a 30B MoE model show that the approach maintains accuracy while achieving up to 1.5× throughput improvement.

**Compliance With Llm Reviewing Policy:**

Affirmed.

**Final Justification:**

I think most of the concerns are addressed. And I will maintain my score.

**Key Questions For Authors:**

1. The paper argues that token-agnostic expert evaluation obscures the contributions of informative tokens. Could the authors provide direct empirical evidence showing how expert importance rankings or pruning outcomes change when critical token selection is used? Clear evidence would strengthen the motivation and technical soundness of the approach.

2. STEP combines critical token selection and loss-aware expert evaluation. Could the authors provide ablations isolating the contribution of each component and report the computational overhead of token selection itself? If both components contribute meaningfully and the overhead is small, it would make the method more convincing.

3. The abstract claims effectiveness across different MoE architectures and model scales. Could the authors clarify how consistent the improvements are across these settings and whether there are cases where the gains are smaller? Demonstrating consistent improvements would strengthen the significance of the work.

4. The paper reports results at 50% expert sparsity. Could the authors provide a broader sparsity–performance trade-off curve to show how model quality changes under different pruning levels? This would help assess the robustness and practical usability of the method.

**Limitations:**

Yes

**Strengths And Weaknesses:**

Strengths

1. The paper addresses an important practical problem—improving the inference efficiency of MoE models—by reducing redundant expert computation, which is highly relevant for large-scale LLM deployment.
2. The proposed STEP framework introduces a token-centric perspective for expert pruning, using token importance estimation to guide pruning decisions, which is a reasonable and intuitive design.
3. Experimental results show consistent throughput improvements while maintaining comparable accuracy, suggesting the approach can improve efficiency with limited impact on model quality.

Weaknesses

1. The experimental evaluation is relatively limited, with experiments conducted on a narrow set of benchmarks and models, leaving questions about generalization to other tasks and MoE architectures.
2. The method description lacks some implementation details, particularly regarding the token importance scoring mechanism and pruning thresholds, which may make reproduction more difficult.
3. The novelty is somewhat incremental, as the work largely builds on existing ideas in token pruning and MoE efficiency optimization rather than introducing a fundamentally new technique.

---

> ### Author Rebuttal · Authors · 2026-03-31
>
> We sincerely appreciate your constructive feedback and recognition of our work's practical relevance. Below we address each concern raised.
>
> **W1: Generalization to Other Tasks and MoE Architectures**
>
> In the original manuscript, we conducted experiments on four model architectures ranging from 7B to 100B parameters, following the evaluation datasets established in prior pruning literature. To further demonstrate generalizability, we have expanded our experiments to include Deepseek-MoE and Ling-mini-2.0, and supplemented our evaluation with additional datasets covering Chinese language tasks, truthfulness metrics, mathematical reasoning, and code generation. The results confirm that our method consistently outperforms the MoNE baseline across these diverse settings.
>
> Detailed results : https://anonymous.4open.science/r/ICML_21218_Rebuttal/generalization_experiments.pdf
>
> ---
>
> **W2: Implementation Details**
>
> We provide the following detailed clarification on our implementation. We extract token importance by hooking the attention module to obtain maximum attention scores across heads, reducing the n×n attention matrix to an n-dimensional vector. We then select the top 50% of tokens (determined via ablation studies) to compute expert evaluation metrics. To facilitate reproduction, we will release our complete implementation code publicly.
>
> ---
>
> **W3: Our Key Contribution**
>
> Our key contribution is not merely applying existing token pruning methods to MoE models, but rather discovering and demonstrating the critical importance of token-level importance in the expert pruning domain—an insight that has been largely overlooked in prior work.
>
> ---
>
> **Q1: Empirical Evidence for Token Selection Impact**
>
> We conducted IoU analysis comparing expert rankings between token-aware and token-agnostic methods across all layers. After applying token selection, the average IoU of the remaining top-32 experts falls below the 80% threshold, with certain layers exhibiting IoU values as low as 50%, demonstrating that critical token selection fundamentally reshapes pruning decisions. More importantly, our method increases the variance of expert score distributions by 1.4× on average, thereby enhancing the SNR of expert scoring.
>
> Figure: https://anonymous.4open.science/r/ICML_21218_Rebuttal/token_selection_impact_on_expert_pruning.pdf
>
> ---
>
> **Q2: Component Ablation and Computational Overhead**
>
> We have extended our ablation studies beyond the 25% pruning rate reported in the original manuscript to include comprehensive component analysis at 50% pruning rate. These results demonstrate that both critical token selection and loss-aware expert evaluation contribute meaningfully to overall performance. Regarding computational overhead, since attention weights can be directly extracted from the model without additional forward passes, the token selection process incurs minimal cost. Specifically, for OLMoE-7A1B, the total overhead including multi-head attention processing, token ranking, and selection amounts to 1.92 seconds out of 58.14 seconds total pruning time, representing only 3.30% overhead. For Qwen3-30A3B, this overhead is 11.01 seconds out of 201.95 seconds, with 5.45% overhead. These negligible overheads confirm the practical efficiency of our approach.
>
> Ablation results: https://anonymous.4open.science/r/ICML_21218_Rebuttal/component_ablation.pdf
>
> ---
>
> **Q3: Consistency Across Architectures and Scales**
>
> Our consistency claims are substantiated by performance improvements across all evaluated datasets, different pruning rates, and multiple model architectures. The method demonstrates stability with respect to calibration dataset length and sample size. We acknowledge that performance gains are smaller when pruning very few experts, which we attribute to the limited impact of removing small numbers of experts on overall model quality. For instance, in Qwen3 when pruning 48 experts, we observe a 0.23% improvement over the best baseline. However, as pruning rates increase shown in Q4, our method maintains consistent advantages, validating its effectiveness in practical deployment scenarios where significant compression is required.
>
> ---
>
> **Q4: Sparsity-Performance Trade-off Curves**
>
> We have extended our analysis beyond the 50% sparsity level reported in the original manuscript. The results now include comprehensive trade-off curves for OLMoE, Moonlight, and Qwen3 models across varying sparsity levels. These curves demonstrate that our method achieves consistent improvements across the entire sparsity spectrum. Notably, the Qwen3 model maintains a 5% performance advantage even at 68.75% sparsity, illustrating the robustness of our approach under aggressive pruning conditions.
>
> Figure: https://anonymous.4open.science/r/ICML_21218_Rebuttal/performance_sparsity_curves.pdf
>
> ---
>
> We believe these additional experiments and clarifications substantially address your concerns and strengthen the empirical foundation of our work.

---

> > ### Author Rebuttal · Reviewer_7TML · 2026-04-03
> >
> > I think most of the concerns are addressed.

---

> > > ### Author Response · Authors · 2026-04-07
> > >
> > > Thank you for your careful review and for noting that most of your concerns have been addressed. We sincerely hope our revisions sufficiently demonstrate the contribution of this work, and we would appreciate your kind consideration of adjusting the score accordingly. Please feel free to let us know if any further clarification is required.

---

### Decision · Program_Chairs · 2026-04-30

**Decision:**

Accept (regular)

**Comment:**

This paper addresses MoE computational and memory overhead by proposing Selective Token-guided Expert Pruning (STEP). The authors identify a critical flaw in existing *token-agnostic* pruning methods: by evaluating expert importance uniformly across all tokens, the crucial signals from highly informative tokens are diluted by the overwhelming majority of less important ones. In order to resolve this, STEP introduces a token-aware framework, which leverages an attention-guided token selection mechanism, combined with a loss-aware expert evaluation and a lightweight bias compensation stage, to permanently remove redundant experts while minimizing information loss. Extensive evaluations show that on the 30B Qwen3 MoE model at 50% sparsity, STEP achieves a 1.5x throughput speedup with the entire pruning process taking less than 10 minutes.

During the rebuttal, the authors provided pseudocode and additional ablations to resolve the core concerns of the majority of the reviewers.

**Addressed Problem**:
- Ablation on Token Selection Metrics: In response to Reviewers C62D and GECX, the authors explored alternative token importance metrics (e.g., gradients, L2 norm, update magnitude) and compared their approach with prompt compression methods like Selective Context and LLMLingua. The additional experiments demonstrated that while gradient-based scoring and other compression methods are effective, the proposed attention-based token selection offers a better balance of empirical performance and computational efficiency.
- Generalization and Extreme Sparsity: In response to Reviewer 7TML's questions regarding model generalization and sparsity trade-offs, the authors expanded their evaluation to include Deepseek-MoE and Ling-mini-2.0. They also provided sparsity-performance curves, showing that STEP maintains a 5% performance advantage over baselines under an aggressive 68.75% sparsity setting.

**Remaining Problem**:
- Distinctions from MoNE and Pruning Paradigm: Reviewer xfix questioned the core design of STEP, arguing it was highly similar to MoNE and lacked innovation. They also considered comparing the online expert skipping mechanism with offline pruning to be seriously unfair. The authors provided pseudo code and static pruning masks to demonstrate that STEP is a static, offline pruning method, and detailed the differences from MoNE in the token selection phase, scoring logic, and global bias optimization. But these responses failed to completely resolve Reviewer xfix's concerns, and Reviewer xfix maintains a negative Weak Reject (3) score.

The AC recommends a Weak Accept, primarily based on the following considerations:
- The majority of the reviewers (Reviewers 7TML, C62D, GECX) expressed recognition of the authors' rebuttal. They agreed that the supplementary ablation studies, computational overhead analysis, and extended validation of model generalization demonstrated the effectiveness and engineering potential of the proposed method, and provided Weak Accepts.
- Reviewer xfix strongly argued that the method is highly similar to existing works (such as MoNE) in terms of core design, with limited marginal novelty, and remained skeptical about the absolute fairness of experimental baseline comparisons even after replication. Reviewer xfix maintains a Weak Reject.

Overall, the AC believes that this work demonstrates practical value in compressing large-scale MoE models and recommends Weak Accept. The AC strongly suggests that the authors comprehensively discuss the mechanistic differences with similar works (such as MoNE) in the final version, and clearly annotate the limitations of its practical deployment.